# The Smart Community: Strategy Layers for a New Sustainable Continental Framework

Vlad Stoicescu [1], Teodora Ioana Bițoiu [2,*] and Cătălin Vrabie [3,*]

1 Bucharest Academy of Economic Studies (ASE)—Business Administration Doctoral School Sustainable Fuels Association (ACS), 010374 Bucharest, Romania
2 Department of Economics and Public Policies, National University of Political Studies and Public Administration (SNSPA), 012244 Bucharest, Romania
3 Faculty of Public Administration, National University of Political Studies and Public Administration (SNSPA), 012244 Bucharest, Romania
* Correspondence: teodora.bitoiu@administratiepublica.eu (T.I.B.); catalin.vrabie@snspa.ro or office@arca.ngo (C.V.); Tel.: +40-723-689-314 (C.V.)

**Abstract:** The topic investigated in this article is a comparison, contrast, and integration effort of European strategies for sustainable development with the evolving market initiatives that are beginning to fuel the fourth industrial revolution. Several regulatory initiatives from continental bodies come into effect to radically change access to finances for business development, based on sustainability goals, and an analysis of the legislation and trends becomes essential for an effective pivot tactic in the face of adversity, as well as change management policies to pre-emptively adapt and perform. The general research question is "what the strategic tools are best employed to overcome the hurdles laid forth by the drastic changes legally required for a sustainable future?" The research methods include a quantitative analysis of norms, regulations, and legislation, including strategic initiatives circulated in the European Union governmental bodies, integrated with qualitative research of the literature. The study finds and draws synergies between national strategies that have recently been drafted or are currently evolving with sustainability-centric initiatives such as the hydrogen initiative, the nuclear initiative, the natural gas initiative, the renewables initiative, the synthetics, and biomass initiative, the ESG initiative, the digital initiative. The findings are to contribute to the business administration field by providing an appropriate image of the organizational design model in the sustainability era, and a strategy framework to build the optimum long-term vision founded on continental regulatory initiatives that have come into effect.

**Keywords:** sustainability; smart; development

## 1. Introduction

The several palpable technological and conceptual evolutions in the true sustainable human habitat, with nature, increased comfort for inhabitants, with redundant workloads and processes lost to automation and digitalization at the core, for a prolonged and more fulfilling livelihood, must all be approached individually and probed for their relevance, and those that stand the test of scientific consensus must become part of a unified theory of the smart community.

The community brand [1] will be carried beyond the city, country, or continental unions, by power of positive example, and stand on its own as a template for horizontal implementation, with each administration improving the model based on the previous experiences and developing a newer, better model while inviting the corporate environment to participate on a stable platform that they can thrive in. It is imperative that any strategy is not treated as a single, one-time effort, and the strategic guidance be amended periodically to reflect new requirements based on innovative technologies or legal amendments. Considering that European legal requirements are prohibitive in acquiring qualitative

consultancy packages [2], statal entities might be deterred from building accurate strategies and regulatory frameworks for sustainable development precisely by the continental regulatory bodies that require them. With such incipient blockages, it is foreseeable that sustainability criteria for public procurement will be second to cost-efficiency. Moreover, the first and foremost foreseeable issue will be the acquisition of an effective–qualitative consultancy package to sketch the initial strategies, meant to influence investments in sustainable development.

Starting from the need for a unified theory, we will restrict the research to the European Union, where burdening bureaucracy is a constraining factor to the evolution of the proposed unified theory, mainly due to a heavy apparatus of functionaries looking to fulfil needs and demands from their own states rather than cooperating for a stronger union [3,4]. The argument is that the E.U. has managed to pioneer the world with a state-of-the-art Emissions Trading System that has no equal in efficiency precisely due to the bureaucratic system, and is academically valid, adding that all member states have participated with knowledge and expertise to develop a rigorous mechanism.

Therefore, we will assume that the E.U. system can provide a valuable framework for the smart community model and base the research on present-day initiatives that the unified theory can be built on. This scientific effort will deliver a quantitative correlation or causation between legislation, normative, and strategic initiatives emitted by the European Union as guides for the smart community unified theory. Qualitative research of literature will build the foundation for a smart community unified theory.

The unified theory is by no means a definitive template to be applied "as is", but it is meant to be a working model for the current stage of sustainable development, an organizational design model based on strategic action, compiled with the instruments at hand as they are emitted by the continental regulatory institutions.

The research aims to focus on infrastructure nodes as conceptual habitats for the production, assembly, sale, distribution, and transportation of critical components for sustainable supply chains, operations, and implementation. Infrastructure nodes are at the centre of enterprise affairs, and the first step towards building a strategy with controllable elements in mind. Analysing the bargaining power of suppliers [5] is the first step in developing a business strategy, and while all suppliers of global corporations might be impossible to assess, at this stage, in a study that aims to result in concrete methods, the supply chain is prioritized as next in line to suppliers and indeed possible to analyse and monitor for conformity with the sustainability requirements.

Infrastructure nodes benefit from adjacent strategies built for long-term development that overlay each other and begin to generate an image of the organization. The Pan-European Corridor (General Masterplan for Transportation), for example, has specific information regarding infrastructure node development [6], with key strategic inputs that are relevant for this research effort, as they overlay with interest for green hydrogen production, transportation, and use. This issue is clearly visible in the National Strategies for Hydrogen that have been emitted by E.U. member states. Moreover, the two former regulatory efforts overlay on the National Intelligent Transportation Systems Strategies of member states that digitally manage the infrastructure node for optimum efficiency.

Considering all these regulatory initiatives are the foundation of new organizational design models by means of strategy formulation, the article is to deliver relevant information regarding such legal initiatives and provide an interpretation of the best possible outcome for the social, public, and economical ecosystems to use for their own sustainable development plans.

The main motivation for writing this article is to propose a unified theory for the development of a smart and sustainable human habitat, specifically in the context of the European Union. The authors argue that current technological and conceptual evolutions in sustainable living must be evaluated individually and that those that pass scientific consensus should be incorporated into the theory. The author also notes that the community

brand will be carried beyond the city, country, or continental unions, by power of positive example, and stand on its own as a template for horizontal implementation.

## 2. Materials and Methods

After reviewing the research outlined in *Sustainability* (issues 2019–2022), *Energies* (issues 2019–2022), *Sustainable Cities and Society* (issues 2019–2022) and *Journal of Ambient Intelligence and Smart Environments* (issues 2019–2022), one can conclude that much of the focus is placed on transportation systems running on alternative fuels in general and little focus is placed on those running on hydrogen alone.

Cremades and Canals [7] researched the future of mobility and provided ten scenarios, concluding that battery-based vehicles will not be the choice of the century. Furthermore, Sterlepper et al. [8] focused on hydrogen internal combustion engines, emphasising that the hydrogen combustion engine will be capable of achieving zero-impact tailpipe emissions, an important statement to be taken into consideration when building policies that are aiming to contribute to the European Green Deal.

Additionally, Candelaresi et al. [9] presented an energy analysis combined with a comparative environmental life cycle assessment aiming to help policy-makers implement well-supported strategies of the use of hydrogen in road transport. Moreover, Wang and Tang [10] systematically analysed previously published papers to evaluate the environmental efficiency of new energy vehicles, concluding that electric vehicles pollute more than actually estimated—a conclusion we also reached in one of our previous studies [11].

Similarly, Orsi [12] started his research on the sustainability of electric vehicles asking, "What about their impacts on land use?". The answer was clearly stated: "energy production may have the largest implications for land use". The author focused on operating costs, charging requirements, and energy production.

Li et al. [13], after a comprehensive analysis of the hydrogen fuel infrastructure investments, presented the risk of ignoring the potential interaction between the adoption process and the speed with which the required transportation infrastructure will become available. The authors emphasised the importance of the hydrogen vehicles and the development of corresponding infrastructure.

For practitioners and leaders of municipalities, Schachtner [14] proposed stable efficiency measures and visionary holisticness of the adoption capability of ambiguous requirements in the public sector. He also designed and architected some of the most important elements of the digital strategies of municipalities as described in the digital initiative in the Results section of the current article. Additionally, Harizaj and Ndreu [15] provided a scenario of living in "Smart Cities and Green World".

The reviewed articles helped drawing the whole picture of what are the current trends in research into the field of transport system fuels with a particular focus on hydrogen. Nevertheless, the cited authors are concerned about greenhouse gases, pollution, and cities' environments along with the positive impact of hydrogen as an alternative fuel.

Quantitative correlation or causation between legislation, normative and strategic initiatives compared, contrasted, and integrated with a qualitative literature review.

Appendix A lists a number of 70 official documents issued by European regulatory bodies, including but not limited to Legislation, Regulations, Directives, Recommendations, Provisions, Initiatives, Strategies, Masterplans, Roadmaps, etc. A study of said documents will come to confirm that European standardisation is highly dense and bureaucratized, but comprehensive as well.

Each of the official documents is either an evolution of previous legislation, through amendments or updates, or a cumulative document that inspires, spans, and evolves into several other bodies of legislation, norms, regulations, strategies, or recommendations (Table 1). A simplified graphical representation of the regulatory documents studied can be found in Figure 1. The research has been developed, over Appendix A, and extended to specific regulatory documents that evolve past (but are closely linked to) environmental regulations. Appendix A research must be regarded as a dynamic wireframe of interde-

pendent instructions for a period-bound assessment of the efforts employed by continental regulatory bodies, that ultimately results in a framework set on pragmatic foundations for the concepts proposed by this research effort.

**Table 1.** Targeted actions as identified in 70 official documents issued by European regulatory bodies (Appendix A).

| Proposal Action Domain | Official Document (as in Appendix A) | Attributes |
|---|---|---|
| Emissions | (1, 2, 3, 4, 5, 6, 7, 8, 9, 10, 12, 13, 14, 15, 16, 17, 19, 20, 21, 25, 26, 28, 30, 31, 35, 36, 37, 40, 42, 43, 44, 45, 46, 49, 50, 51, 55, 56, 57, 59, 68) | Emissions trading for roads<br>Emissions trading for buildings<br>$CO_2$ emissions standards for cars and vans<br>EU Emissions Trading System<br>Aviation Initiative<br>Carbon Adjustment Mechanism |
| Social Climate | (*1, 14, 15, 16, 17, 20*, 25, *29*, 31, *42, 43, 44, 45, 46, 49, 50, 52, 53, 54, 58, 60*) | Funding<br>Sharing Regulation |
| Land Use | (*2, 19, 20, 25, 31, 55, 56*) | Land Use Change |
| Forests | (3, 5, 6, 7, *14, 15, 17, 19, 20*, 25, 30, *31, 55, 56*) | Forestry Regulation<br>EU Forest Strategy |
| Energy | (1, 4, 6, 8, 9, 10, 11, 12, 13, 14, 15, 16, 17, 18, 19, 20, 21, 22, 23, 24, 25, 27, 28, *31*, 32, 33, 34, *38*, 42, 43, 44, 45, 46, 49, 50, 51, 55, 56, 57, 58, 59, 60, 68, 68, 69, 70) | Aviation Initiative<br>Maritime Initiative<br>Energy Taxation<br>Renewable Energy Directive<br>Energy Efficiency Directive |
| Renewable | (4, 6, 8, 9, 10, 11, 12, 13, 14, 15, 16, 17, 18, 19, 20, 21, 22, 23, 24, 25, 27, 28, 31, 35, 36, 37, 38, 40, 42, 43, 44, 45, 46, 49, 50, 51, 55, 56, 58, 59, 67, 69) | Aviation Initiative<br>Renewable Energy Directive |
| Alternative Fuels | (4, 6, 8, 9, 10, 12, 13, 14, 15, 16, 17, 19, 20, 25, 31, 35, 36, 37, 38, 40, 51, 55, 56, 57, 58, 59, 60, 68, *70*) | Infrastructure Regulation |
| Standards | (1, 2, 8, 9, 10, 12, 13, *14, 15, 16, 17, 19, 20*, 22, 25, 26, 27, 30, 31, 35, 36, 37, 38, *39*, 40, 42, 43, 44, 45, 46, 49, *50, 51, 52, 53, 54, 55, 56, 58*, 66, 70) | $CO_2$ emissions standards for cars and vans |

The documents in *italic* do not mention the research concept specifically; however, they do contain information or discuss actions and strategies related to it. Some of the studied documents (41, 47, 48, and 61–65) are hardly linked with the research concepts; however, they are important in creating the framework for their development (e.g., funding programs).

Regulation (EU) No 1315/2013 is a recommended pivot point for developing a unified theory for the smart community, as it lists the "Union guidelines for development of the trans-European transport network". The scientific literature is abundant regarding correlations between infrastructure and community development, with industry and society developing in unison on strategic infrastructure nodes since humanity has begun to evolve [16–18].

Several European regulation documents have been emitted and have been studied to be included in Appendix A that begin to build on the unified strategy theory by consolidating communication networks to link and expand the community principles [19], evolving past physical realms of societal dynamics by data-driven networks [20,21]. The Digital Single Market Strategy, heavily subsidised for implementation by several economic packages such as Horizon Europe and The Recovery and Resilience Program, is evidence of the Union's efforts to consolidate continental and global communities via digital instruments.

Under regulatory body guidelines, Intelligent Transportation Systems (ITS) National Strategies are beginning to take shape in nation states, with a first-and-foremost purpose to reduce waste in supply chains by reducing errors in transportation networks to a minimum, with a direct influence in reduced carbon emissions [22]. A secondary result of ITS evolution is a drastic reduction in transportation accidents by remote and autonomous management

of people and cargo logistics [23] with a correlation between transportation infrastructure nodes and digital infrastructure nodes required for broader data transfers that connect transportation machinery to command centres, that insure continuity in automated facility management, that process data and metadata for ever-greater efficiency on routes, and that provide the best financial option correlated to the most environmentally sound offer from provider to client [24].

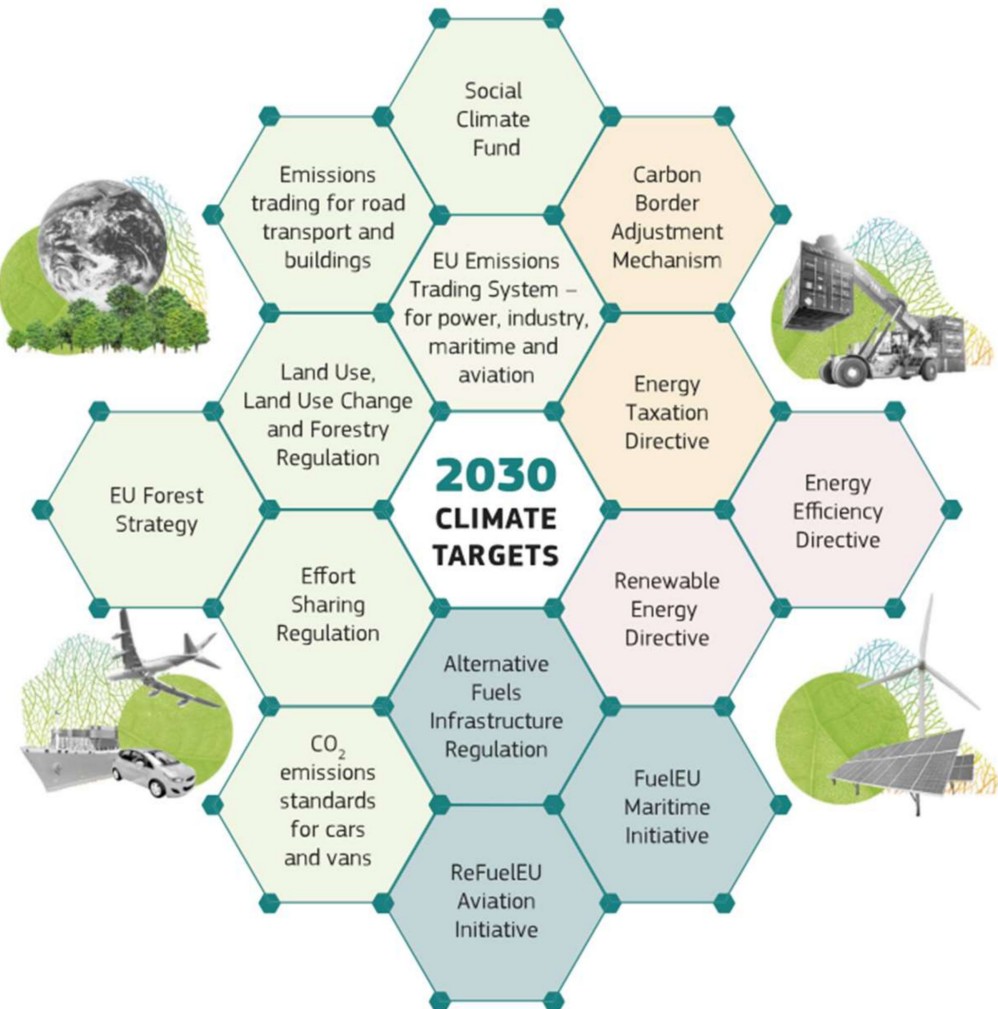

**Figure 1.** The legislative proposals made by the Commission will provide the necessary tools for the deep and just transformation of the EU's economy envisaged by the European Green Deal. Source: https://op.europa.eu/webpub/com/general-report-2021/en/ (accessed on 10 December 2022).

Environmental aid has risen to the centre of attention with the EU Emissions Trading System (ETS), that includes ambitions to reduce 55% greenhouse gas emissions (GHG) from the Union by 2030, calculated in comparison with 1990 archived registries. A more recent agreement on EU ETS stipulates that the reduction increases drastically in ambition, as the new goals are to reduce GHG by 62% to 2030, reported on 2005 figures [25].

The Carbon Border Adjustment Mechanism levels out the global supply that is openly dealt on European markets, and reduces well-known discriminating practices of outsourcing carbon-intensive production to countries where environmental concern is at a minimum and damaging industrial practices are still the norm [26]. Draconic fiscal measures come into effect, forcing importers to calculate both the environmental impact of production and the supply chain effects on the environment and pay mitigation fees [27].

The EU ETS and the foreseeable ETS2 package [28], which covers issues of caps and reductions in GHG, rules for market stability, extensions on maritime transportation, a

separate regulation and system for buildings and road transport, supplements on the Innovation and Modernisation Funds, and improved regulations on ETS revenue management, will have direct effects on both environmental affairs, directly reflected in an improved lifestyle for smart communities and on a global scale, influencing EU external suppliers and importers to act more responsibly in trade with fiscal penalties insured otherwise.

The EU ETS, CBAM, and RefuelEU package have a direct influence on external suppliers, forcing them to adjust their production quality and equal the EU standard in order to comply with imports standards on the continent, resulting in a positive global impact. Similar to the "employer branding" concept and strategies [29], EU regulatory bodies and corporations that comply to legislation, norms, and regulations will have a direct influence on the betterment of global communities via their supply chain and global reach, an effect nuanced in the academic literature as "community branding" [1]. A simplified explanation of the conceptual framework resides in the power of corporate affairs to produce positive effects remotely, by enforcing only those standards that directly benefit both production communities they employ and consumer communities they cater to. With the new regulations that come into effect on European premises, economical agents are mandated to enforce minimum standards for equitable production and consumerism.

Non-financial reporting (NFR) requirements gradually come into effect [30], with Appendix A listing the sinuous consultative and legislative efforts that gradually grew into the European Banking Authority Roadmap on Sustainable Finance [31]. Figure 2 lists a series of recommendations meant to aid banking institutions in financing solely those ESG-compliant investments, paving the path to a foreseeable filter on finances for enterprises that are lacking in their efforts to operate in correlation with the UN Sustainable Development Goals (SDGs), as adopted by the European Union via the Paris Agreement [32].

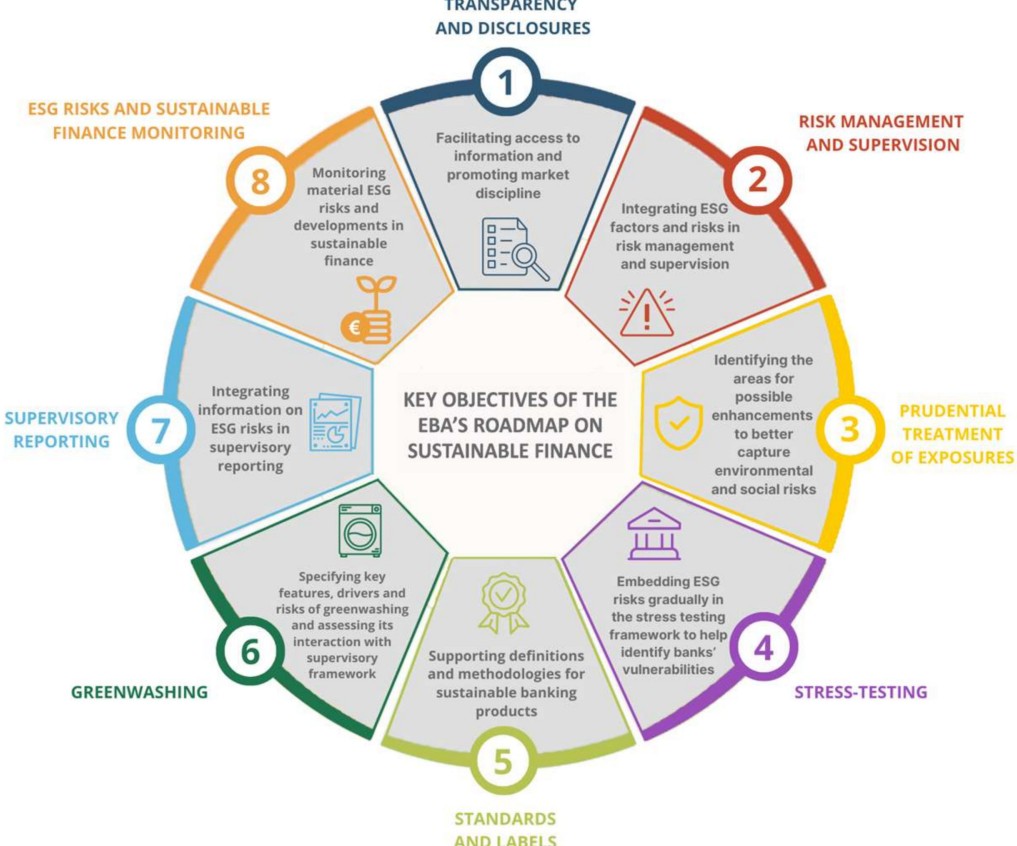

**Figure 2.** Key objectives of the EBA's Roadmap on Sustainable Finance. Source: https://www.eba.europa.eu/eba-publishes-its-roadmap-sustainable-finance (accessed on 10 December 2022).

The SDGs rapidly influenced the development of a global green bond market, which corporations gradually grew to use to their advantage, with some knowing and others ignorant of the dealings that produce little to no positive effect on the betterment of the smart global community [33], an effect that came to be known as "Greenwashing", to be blamed on, amongst others, asymmetric correlations between corporate finances and erroneously calculated time-bound projects [34].

The SDG criteria is by no means flawed, and it is comprehensive enough to ensure an equitable economic environment, with requirements for 17 goals, as visible in Figure 3. However, the European Union further enforces the goals with the Commission guidelines on NFR, and several guides, norms, and recommendations that have influenced or are evolved from the guidelines, as listed in Appendix A.

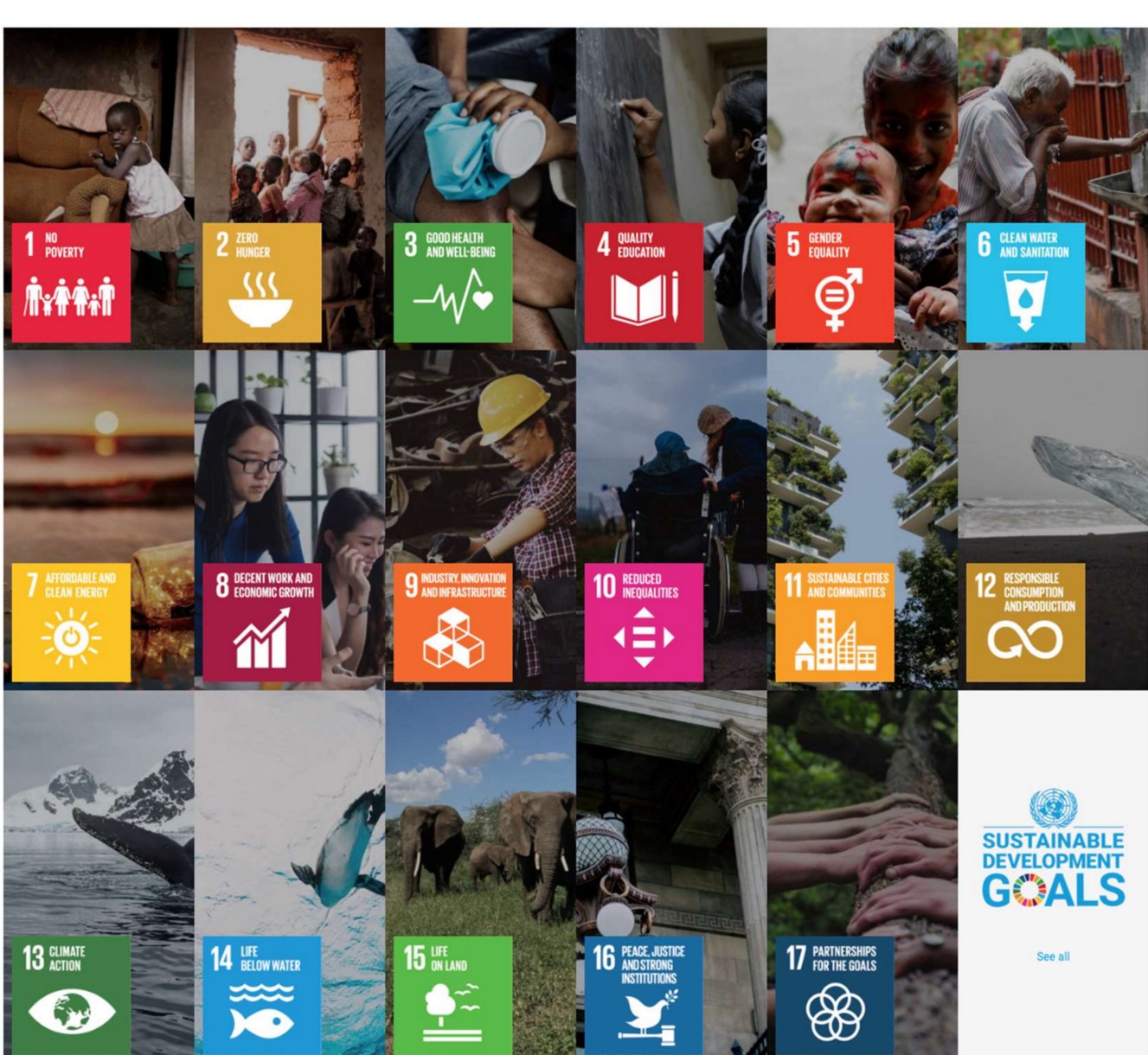

**Figure 3.** UN SGDs. Source https://sdgs.un.org/goals (accessed on 10 December 2022).

The guidelines, as evolved from the Directive 2014/95/EU, come to stimulate European organisations to report on compliance efforts with the SDGs, but full reports have yet to appear [35], as corporations manifest an interest in presenting their efforts to the market as a P.R. mechanism, but are still more inclined towards substantial efforts in reducing OPEX by outsourcing to less sustainability-demanding production sites. While NFR guides have been optional and improper reporting should have acted as a red flag on a vigilant market, the EBA Roadmap is a sign that further measures will come into effect and financing options for public interest companies will be restricted without complete disclosure and compliance in the near future.

The EBA, European Insurance and Occupational Pensions Authority (EIOPA), and European Securities and Markets Authority (ESMA) report on undue short-term pressure from the financial sector on corporations; all agree on the need to curb sustainability failures by influencing a shift from the short-term corporate vision and dynamics to long-term strategies with sustainability commitments [36]. All reports listed in Appendix A lead to a common hypothesis of restricted finances and financing options for organizations that fail to report or plan with SDGs, but further regulation is expected to enforce sustainability as a common standard for economic dynamics. A reform is imminent, as the current global crises would not allow lobbying to continue prevailing over social interest on sustainability [37].

Corporate carbon disclosure will require thorough logging of emitting activities. Moreover, SDG disclosure is gradually becoming mandatory, if not for regulatory reasons, then to satisfy ever-increasing market demand for sustainable products and services. Monitoring and reporting activities both demand ESG-focused staff to backlog and maintain an updated corporate ledger, all resource-intensive issues that must shape the organisation design via its quality system [38–40]. Total Quality Management (TQM) and ESG become synergic in the design management principles for organisations that opt for a gradual compliance strategy that is drawn out to minimise the stress on operations in the transition period.

The ISO 14064-1:2018 standard is a first step in TQM and GHG correlation, and it paves the path to process automation, with procedures set in place for digital wireframing over quality measures to compile software solutions for the mechanical accounting of emissions [41]. GHG accounting paves the way for fully automated ESG monitoring and reporting, with ISO 26000 already available for process mapping [42], but restricted to CSR measures. ISO 14007 (Determining Environmental Costs and Benefits) and ISO 14008 (Monetary Valuation of Environmental Impacts and Aspects) are being set in place to standardise the "book value of nature" for a cost/benefit analysis that permanently allows corporations to benchmark against SDGs [43].

The forthcoming ISO 14030 (green bonds) is set to deliver standardisation on several aspects related to green bond emission and acquisition [44], while ISO 14097 (Climate-Related Metrics for the Finance Sector) provides standard regulation and frameworks for financial activities related to carbon reduction, with ISO 14007 and 14008 providing frameworks for measuring environmental costs and benefits or valuations on sustainability [45].

Relevant standards are in place for full process automation on ESG, resulting in guides for digital platforms to be built on, for complete and complex accounting starting from GHG and evolving into full SDG evaluation. Moreover, such platforms can be easily confronted with potential investments for offsets and compensation, based on appropriate projects that best suit organisation needs and smart community development at the same time.

ReFuel EU [46] begins to elaborate the subject of alternative transportation fuels, as it is meant to amend the Renewable Energy Directive (RED II) and develop sustainable alternatives to fossil fuels for large consumers in all transportation industries, with a clear and concise focus on aviation and maritime (river) transport and a revision to the directive on alternative fuels infrastructure deployment. A majority of the alternative fuels proposed to compete with hydrocarbon-based catalysts are either derivatives of hydrogen, require hydrogen for production, or are various hydrogen-based technologies;

the European Commission released a supranational strategy for hydrogen deployment, including all required aspects for gradual implementation [47,48].

Along with decarbonisation ambitions, ReFuel EU is dense in solutions for an equitable and equal playing field on the global logistics and transportation market, with obligations for transporters to accept gradually increasing quotas of European-originating sustainable fuels, thus eliminating unfair competition from transporters that have access to cheap fuelling solutions [49]. Fuels are the main operational expenditure in transportation, and drastic variations in price, from continent to continent, invariably result in unfair competition and discrimination against European carriers, an issue that is bundled in the set of environmental solutions under ReFuel EU.

Fit for 55 includes the COM/2021/559—Alternative Fuels Infrastructure Regulation which delivers guides for alternative fuels infrastructure development and financing mechanisms to incentivise market development [50]. However, with an incipient industry that is regulated for industrial hydrogen production, several issues arise with regard to green hydrogen synthesis, transportation, distribution, and delivery to vehicles, that member states must regulate before widespread adoption into transportation. An evolution of hydrogen, as a competitive transportation fuel, compared to hydrocarbon based fuels, requires extensive networks of on-site production as transportation is cumbersome and cost-inefficient; renewable energy resources; carbon capture technology; and a wide-scale adoption expanded into networks for economies of scale to foster the development of "Hydrogen Valleys" [51].

The hydrogen valley is a multi-modal industrial site that facilitates the production, logistics, delivery, and utilisation of hydrogen as a fuel for industrial production and transportation. Steam Methane Reform (SMR) production of hydrogen is still being considered, as long as the technology (which requires natural gas burning) is fitted with carbon capture devices; however, electrolysis fuelled by renewable energy is the optimal approach [52]. According to EU Taxonomy Regulation updates, nuclear energy and natural gas can be considered renewables under certain conditions [53], paving the way to fuel a hydrogen valley network across Europe and consolidating an incipient industry to compete with conventional fuels.

Several issues must be resolved to eliminate hurdles on the path to implementing the regulatory initiatives listed, most stemming from the Union's overprotective habits that, paradoxically, are contradicting an industry-supported logarithmic evolution rate of technological advancements inspired by the same regulatory initiatives. Strategic public procurement is legally prohibitive with respect to favouring the most financially efficient offer on the market [54], an issue that some countries respect with great diligence to avoid corruption charges [55]. Even with the official "Communication on public procurement for a better environment" and the combined Commission Staff Working Document, Green Public Procurement (GPP) is still avoided in Eastern European countries due to the criteria being recommended rather than imposed and the main procurement legal instruction continuing to favour a lesser product, in environmental terms, due to the cost/benefit analysis pointing towards it.

However, the EU GPP norms have been found in efficient legislation chapters, such as the Energy Performance Directive (2010), Energy Star Regulation (2008), Clean Vehicles Directive (2009), and Energy Efficiency Directive (2012) [56], findings that result in the assumption that recommendations will not be effective without subsequent legislation to enforce sustainability-driven initiatives, driving the nation states to comply and adapt.

## 3. Results

A comprehensive set of regulatory materials have been compared, contrasted, and integrated into a revealing review of the scientific literature, for a combined quantitative and qualitative analysis required to develop a unified theory of organizational design models considering national and continental strategic initiatives for sustainable development.

As relevant in Appendix A, legal and regulatory initiatives are interdependent and independent at the same time, having minimal common ground but forming a regulatory framework that extends to cover more issues than any other effective body of legislation. This is relevant to the bureaucratic nature of the European Union, and observed superficially, it may appear fuzzy and incoherent. However, a profound analysis reveals several components that can be quantified as palpable results for the organizational design model. For the purposes of this study, the results will be divided into seven subheadings extracted from the combinations, correlations, and extrapolation of continental strategic initiatives, supported by regulatory frameworks and transported into member state strategies for the continental cohesion of industrial standards.

### 3.1. The Hydrogen Initiative

The research supports a theory of continental effort to develop an alternative to conventional fuels, and break the monopoly that hydrocarbons have in industrial dynamics and mobility. Hydrogen is sustained in multiple regulatory initiatives (i.e., 4, 6, 8, 9, 10, 12, 13, 14, 15, 16, 17, 19, 20, 21, 27, 31, 32, 3, 34, 35, 36, 37, 38, 40, 60, 68 and 70 in Appendix A) and financing vehicles to evolve as the main competitor to petrol and gases. In our previous study [11], we stated that transportation will include hydrogen-powered vehicles as an alternative to internal combustion and electric engines.

Hydrogen production requires investments in hydrolysis plants that, themselves, require renewable energy to function in the designed standards. Renewable energy is defined in legislation, but the definition is subject to change, as nuclear energy and natural gas burning with carbon capture technology have been included in the definition to supplement hydroelectric production, photovoltaics, wind power, biomass, and geothermal energy.

Green (renewable-energy-fuelled) hydrogen production is strategically placed in hydrogen valleys, industrially dense regions that evolve around fuel production. Hydrogen valleys are meant to host industrial processes, multi-modal transportation hubs for passengers, cargo, and fuel transfers to demanding recipients. The industrial complex is in proximity to human-resource-rich communities that thrive based on the economic dynamics resulting from hydrogen-centric sustainable development.

In our analysis, there is a significant medium-positive relationship between the number of documents mentioning hydrogen as an alternative fuel and all the EU initiatives on energy ($r(25) = 0.463$, $p = 0.015$) which shows the importance of it in the actual legislation framework.

The hydrogen initiative is an overlay of several sustainable development strategies, including but not limited to the hydrogen strategy, the energy strategy, the transportation masterplan, the intelligent transportation systems strategy, the smart city strategy, and the economic and commerce strategy of a European region or country, adapted from the continental framework provided.

### 3.2. The Nuclear Initiative

Hydrogen demand is forcing a reconsideration of hydro, photovoltaic, and wind power as the main renewable energy supplies, due to their sporadic availability and lack of technological advancements in energy storage.

Advancements in nuclear technology come to sustain a theory of regional energetic independence fuelled by small modular reactors (SMR) or micro-modular reactors (MMR), able to act as the main source of energy for hydrogen valleys, industrial sites or even regions, cities, and nations.

Energy strategies are being amended to include nuclear sources of grid surplus, together with a component of independently fuelled regions of interest, only connected to the grid for emergencies, as seen in some of the studied documents (i.e., 4, 6, 14, 16, 17, 20, 32, 33, 34, 55, 56).

European regulations begin to relax around nuclear technology, signalling a shift towards a widespread adoption of continental energy independence, an all-too-present

narrative in strategic initiatives that have been studied. However, the correlation between the number of documents mentioning nuclear energies among all the initiatives on energies is non-significant small positive (r(9) = 0.158, *p* = 0.643).

### 3.3. The Natural Gas Initiative

Legislation, strategies, and regulation define natural gas as a transition fuel, with hydrogen being the desired goal. Natural gas strategies include transportation infrastructure development, an essential element in the development of hydrogen networks, as European gas ducts are under mandate to transport a mix of gas and an ever-increasing quota of hydrogen.

All natural gas consumers can be retrofitted to burn hydrogen, drastically reducing GHG emissions. European strategies, influencing national strategies mandated by the Recovery and Resilience Facility, point to the direction of gas infrastructure modernisation and hydrogen production for transportation via ducts for multiple-use cases, spanning from central heating in thermal plants to extracting from the mix and supplying vehicle-fuelling stations.

There are specific requirements to reduce the production of hydrogen from steam methane reform, a gas-intensive process, and shift towards hydrogen production from electrolysis, as SMR hydrogen is regarded as redundant for a strategy that ultimately achieves the transportation of pure hydrogen on the European duct infrastructure.

### 3.4. The Renewables Initiative

Part of the energy strategy, renewable energy production technology continues to evolve, capturing more and more of nature's potential to fuel industries, mobility, and communities. A sustained innovation rhythm will result in better yields from harnessing elemental energy, on a long-term prognosis.

All strategic initiatives encourage research and development for renewable energy production; there is a significant large positive relationship between renewable energy documents and EU energy initiatives, (r(40) = 0.976, *p* < 0.001)). However, the current capacities are not sufficient for energy independence targets (energetic strategies) or for the hydrogen quotas needed to be produced by electrolysis (hydrogen strategies).

### 3.5. The Synthetics and Biomass Initiative

Synthetic fuels and biomass are circular-economy-driven efforts that aid a transition to 0 emissions by fuelling industries and mobility without the need for the technological overhaul of machineries, as these NET-0-GHG-emitting fuels can be loaded into currently operating technology.

All strategies and regulatory documents treat synthetics and biomass as transition elements, admitting they still have an impact on the environment but regarding them as a less harmful solution accepted for economic stability on the path to technological evolution in renewables and hydrogen. Market stability is a very important factor in strategy elaboration, and synthetic fuels or biomass curb a need for the complete re-engineering of fuel consumers, allowing for a gradual transition based on existing technologies.

Moreover, strategies are calculated with life cycle costs in mind, ensuring a transition period where current investments are amortised and/or reused by modernisation, with the Union encouraging circular economy tactics for retrofitting machinery to be fuelled by transition solutions, with clearly calculated goals for a long-term transition to new technologies.

### 3.6. The ESG Initiative

ESG is the most documented initiative in implemented legislation and relevant strategies, as it is regarded top-down as a market regulation and bottom-up as a stimulation package. All ESG-related mechanisms are meant to develop a self-sustained system that penalizes non-compliant agents to the benefit of proactive market players.

ESG strategic foresight starts from environmental action, with the E.U. ETS enforcing carbon accounting for economic agents, restricting access to financing for GHG intensive market operations that fail to adopt corrective action. The CBAM extends the E.U. ETS to global markets, forcing economic agents to comply with E.U. regulations and sustain continental strategies by acting equitably. Moreover, CBAM and ETS reduce unfair competition from countries that enforce lesser standards than those mandated by the European Union.

By penalising disparages between international ESG accounts of organisations, the Union builds upon the principle of a global smart community brand, where all economic agents are forced to comply either by paying to comply or transparently reporting on ESG compliance for financial burden reduction on imports to the E.U. Economic agents are forced to disclose compliance efforts in non-financial reports, that are now mandatory for access to finances, and there is a consensus amongst European regulatory bodies that short-term performance indicators for economic growth are not considered business-planning best practices anymore, raising the demand for long-term, sustainable development goals in finance requests.

The standardisation of sustainability measurements is taking shape in organisational design, with TQM delivering instruments for precise evaluation and proactive corrective action, paving the way for automated measuring mechanisms to be developed and implemented in a digital market for ESG. This results in automated compensation with sustainable projects developed precisely to absorb penalties applied to organisations that have yet to fully comply with ESG requirements, projects that positively influence the global community and eliminate the harm caused by failing to initially comply.

More and more sites that absorb finances from this compensation mechanism can be developed, for the benefit of communities worldwide, sites that must promote SDG principles as well and compete between each other by complying with said norms, where partially complying will result in fractured financing and full compliance will result in immediate funding.

ESG quantified in TQM and automated in digital platforms is a logical evolution towards a transparent market where greenwashing practices can be flagged and eliminated to the benefit of both organisations that are proactive in sustainable development and communities that absorb compensation finances and develop in sustainable principles. This results in a self-governed society framework where state entities take a step back from fiscal intermediation and allow for the economic environment to contribute directly to self-developing smart communities.

All strategic initiatives are geared towards the self-administration organisational design model, at a continental level, and the scientific literature brings evidence of continental regulation meant to eliminate nation-state-lobbying practices in favour of a fully automated fiscal compensation mechanism, where human intervention for short-term illicit benefits becomes impossible. Strategy formulation does not directly address the issues of continued correction of illicit economic activity, but all strategies resolve immoral practices with systemic correction.

### 3.7. The Digital Initiative

Full automation for digital governance of all social and entrepreneurial dynamics is prioritised by the E.U. in several strategic initiatives. From public administration digitalisation to digital collaboration platforms between public and central administrations, where economic agents can immediately connect for rapid business deployment, to intelligent transportation systems and ESG platforms, including digital twins for infrastructure and smart cities, all systems are meant to reduce waste and improve communications between supply and demand.

A direct result of digital integration strategies is the environmental assessment of all undergoing administrative or economic processes. Having the ability to raise red flags in heavy-consuming or GHG-emitting processes will result in immediate corrective action.

Moreover, a transparent administrative process and corporate evaluation platform will directly benefit the smart community that uses integrated platforms to its advantage.

Continental strategic initiatives and financing mechanisms stimulate the development of digital platforms for both economic agents and public administrations, with several R&D resources for complete integration.

### 3.8. Study Limitation

Our study is focusing on the European Union and its bureaucratic system as a framework for the development of the proposed unified theory for a sustainable human habitat. The research only looks into the initiatives and guidelines emitted by the European Union for building the theory and does not consider other global or regional perspectives. Additionally, the research also prioritizes on infrastructure nodes as the focal point for sustainable supply chain, operations, and implementation, which may not be inclusive of other important aspects of sustainable development. The study only analyses the bargaining power of suppliers in the European supply chain and not all suppliers of global corporations which may limit the generalization of the findings.

## 4. Discussion

This comprehensive study of strategic initiatives, regulatory documents, and the scientific literature results in valuable conclusions and action paths extracted from an otherwise fuzzy distribution of burdening documentation, a fact especially relevant in Appendix A, that lists normative documents in order and logic that regulatory bodies meant for them to be studied. Various regulations come together in continental strategies that member states must adapt to their respective mechanisms, mimicking independence in self-governance but ultimately resulting in continental systems.

A fair assumption is that of the intended apparent regulatory chaos to avoid bottom-up resilience to change and gradually introduce top-down innovative mechanisms with the support of economic agents and willing communities. A metaphoric parallel to the construction industry would support the theory of a stable building being raised in a chaotic site, filled with state-of-the-art technology and the best engineers available for the job, ultimately coming together in an architectural masterpiece.

This study identifies the foundational pillars of said construction, all relevant and described in the seven initiatives in the Results section. Hydrogen is meant to ultimately fuel continental dynamics and industries, with investments in hydrogen infrastructure developing new economic avenues and centring continental finances against inflation, to the advantage of the currency. Hydrogen reshapes transportation networks and transportation vehicles, a finding supported by market evolutions worldwide.

However, hydrogen production is energy-intensive and requires advancements in nuclear technologies. The E.U. has approved nuclear energy as a renewable resource for tertiary production. Green hydrogen can be produced by electrolysis fuelled from nuclear energy, resolving the issue of hydrogen production inertia on the continent by introducing new nuclear technologies, already visible with Nuscale's Small Modular Reactor fast-tracked introduction on the Romanian energy market and several other initiatives relevant in Poland.

There is not sufficient evolution yet on the path to green hydrogen as an alternative to conventional fuels, resulting in a drastic need for a transition strategy, drawn up in the short term and fuelled by natural gases and circular economy projects. Natural gas infrastructure requirements are building the hydrogen transportation networks, as gas pipes are already mandated to transport a hydrogen mix in them for gradual adjustments to full hydrogen transportation, as the target is set in strategic documents. To meet the targets, industrial-machinery-retrofitting options begin to take shape in circular economy strategies, reusing transportation and production machines with sustainable fuels, either synthetic, biomass-based, or natural gas until transition goals are met.

Renewables remain a long-term solution and several strategies continue to encourage the development of technologies able to capture natural elements and transform them into energy resources. However, a more pragmatic approach to renewables begins to take shape, as their efficiency has yet to improve for renewables to be considered as a stable alternative source of energy.

ESG is the most important component in continental strategy formulation, as it is a framework for self-governance between the economic ecosystem and communities that stand to benefit from a new financial mechanism, built to eliminate political administrators from centralising and distributing fiscal instruments. With several legislation initiatives that regulate organisational behaviour, forcing them to adopt sustainable operation strategies in the entirety of their supply chains, the E.U. has built a global framework for sustainability that does not discriminate against complying agents, as has previously been the case.

However, for ESG strategies to take shape, the widespread adoption of digital infrastructure in administration and industrial operations must become a reality. TQM is rapidly adapting with new standards to meet regulation and strategic foresight, mapping processes that can rapidly be implemented into independent digital platforms, that can be unified on global management systems.

## 5. Conclusions

The smart community is shaped by the economic dynamics that define it. Economic agents will reshape their organisational design to cater directly to their community, a source for human and financial resources, whose evolution is defined by the community branding concept. Corporate interaction is gradually shifting from elected administrators to smart communities via existing digital infrastructure. Economic agents are already free to pool the markets where they operate on social networks, where they are beginning to realise they are held accountable for both the positive and, especially, the negative influence they have on communities.

The European Union is proactive in shaping an interdependent system for self-governance between economic agents and smart communities, where accountability is enforced by the transparency regulations set in place. A strategy for consolidated sustainable development is shaped, with flaws, but regulatory bodies have proven they are able to react and adapt to swiftly correct errors on the path to a sustainable future.

European strategies are being built to support this path with sustainable fuels for industries and mobility, consolidating all options into long-term evolution plans with set goals. Meeting the goal requires the same levels of perseverance from regulatory bodies, and an exponential growth in technological adoption will set in motion digital control mechanisms. More research is needed to develop a perfect platform for ESG dynamics between economic entities and smart communities, where corporations are allowed to directly contribute to the betterment of the communities they influence.

As a final conclusion: the authors state that it is important to consider that European legal requirements are thus far prohibitive in acquiring qualitative consultancy packages, which may deter entities from building accurate strategies and regulatory frameworks for sustainable development.

**Author Contributions:** Conceptualization, V.S. and C.V.; methodology, V.S. and T.I.B.; software, C.V.; validation, C.V.; formal analysis, V.S.; investigation, T.I.B.; resources, V.S., T.I.B. and C.V.; data curation, C.V.; writing—original draft preparation, V.S.; writing—review and editing, C.V.; visualization, T.I.B.; supervision, C.V.; project administration, C.V.; funding acquisition, V.S., T.I.B. and C.V. All authors have read and agreed to the published version of the manuscript.

**Funding:** This paper was co-financed by MDPI on the framework of Smart Cities International Conference, The Bucharest University of Economic Studies during the PhD program, and individual contributions from the authors.

**Data Availability Statement:** Data available in a publicly accessible repository: The data presented in this study are openly available in public repositories, as relevant in the reference index. Data available

in a publicly accessible repository that does not issue DOIs: Publicly available datasets were analyzed in this study. This data can be found on regulatory bodies web-sites (European Union, Council, Public Institutions, United Nations, etc.). 3rd Party Data: Restrictions apply to the availability of these data. Data was obtained from global regulatory bodies and are available at the linked addresses in Appendix A, corresponding to each disseminated dataset. Data is contained within the article: Data presented in this study are available in Appendix A.

**Conflicts of Interest:** The authors declare no conflict of interest.

## Appendix A

1. Directive of the European Parliament and of the Council amending Directive 2013/34/EU, Directive 2004/109/EC, Directive 2006/43/EC and Regulation (EU) No 537/2014, as regards corporate sustainability reporting

   *"The NFRD applies to large public-interest entities with an average number of employees in excess of 500, and to public-interest entities that are parent companies of a large group with an average number of employees in excess of 500 on a consolidated basis. 2 The NFRD exempts subsidiaries from its reporting obligations if their parent company does the reporting for the whole group, including the subsidiaries. Approximately 11,700 companies are subject to the reporting requirements of the NFRD 3."*

   https://eur-lex.europa.eu/legal-content/EN/TXT/?uri=CELEX:52021PC0189 (accessed on 20 December 2022)

2. Corporate sustainability reporting—Directive 2014/95/EU of the European Parliament and of the Council

   *"of 22 October 2014, amending Directive 2013/34/EU as regards disclosure of non-financial and diversity information by certain large undertakings and groups"*

   https://eur-lex.europa.eu/legal-content/EN/TXT/?uri=CELEX%3A32014L0095 (accessed on 20 December 2022)

3. The European Green Deal

   *"The EU is fighting climate change through ambitious policies at home and close cooperation with international partners."*

   https://ec.europa.eu/info/strategy/priorities-2019-2024/european-green-deal_en (accessed on 20 December 2022)

4. REPowerEU: affordable, secure and sustainable energy for Europe

   *"In response to the hardships and global energy market disruption caused by Russia's invasion of Ukraine, the European Commission presented the REPowerEU Plan."*

   https://ec.europa.eu/info/strategy/priorities-2019-2024/european-green-deal/repowereu-affordable-secure-and-sustainable-energy-europe_en (accessed on 20 December 2022)

5. Climate action and the Green Deal

   *"The European Green Deal aims to make Europe climate neutral by 2050."*

   https://ec.europa.eu/info/strategy/priorities-2019-2024/european-green-deal/climate-action-and-green-deal_en (accessed on 20 December 2022)

6. Regulation (EU) 2021/1119 of the European Parliament and of the Council of 30 June 2021

   *"Establishing the framework for achieving climate neutrality and amending Regulations (EC) No 401/2009 and (EU) 2018/1999 ('European Climate Law')"*

   https://eur-lex.europa.eu/legal-content/EN/TXT/?uri=CELEX:32021R1119circy (accessed on 20 December 2022)

7. Communication from the Commission to the European Parliament, the Council, the European Economic and Social Committee and the Committee of the Regions-A new Circular Economy Action Plan For a cleaner and more competitive Europe

*"As half of total greenhouse gas emissions and more than 90% of biodiversity loss and water stress come from resource extraction and processing, the European Green Deal launched a concerted strategy for a climate-neutral, resource-efficient and competitive economy. Scaling up the circular economy from front-runners to the mainstream economic players will make a decisive contribution to achieving climate neutrality by 2050 and decoupling economic growth from resource use, while ensuring the long-term competitiveness of the EU and leaving no one behind."*

https://www.eea.europa.eu/policy-documents/communication-from-the-commission-to-1 (accessed on 20 December 2022)

8. Directive 2009/125/EC of the European Parliament and of the Council of 21 October 2009 establishing a framework for the setting of ecodesign requirements for energy-related products (recast) (Text with EEA relevance)

*"This Directive establishes a framework for the setting of Community ecodesign requirements for energy-related products with the aim of ensuring the free movement of such products within the internal market. "*

*"This Directive provides for the setting of requirements which the energy-related products covered by implementing measures must fulfil in order to be placed on the market and/or put into service. It contributes to sustainable development by increasing energy efficiency and the level of protection of the environment, while at the same time increasing the security of the energy supply."*

*"This Directive shall not apply to means of transport for persons or goods."*

*"This Directive and the implementing measures adopted pursuant thereto shall be without prejudice to Community waste management legislation and Community chemicals legislation, including Community legislation on fluorinated greenhouse gases."*

https://eur-lex.europa.eu/legal-content/EN/ALL/?uri=CELEX%3A32009L0125 (accessed on 20 December 2022)

9. EU Ecolabel

*"The official European Union voluntary label for environmental excellence. Established in 1992 and recognised across Europe and worldwide, the EU Ecolabel certifies products with a guaranteed, independently-verified low environmental impact. To be awarded the EU Ecolabel, goods and services should meet high environmental standards throughout their entire life cycle: from raw material extraction through production and distribution to disposal. The label also encourages companies to develop innovative products that are durable, easy to repair and recyclable."*

https://environment.ec.europa.eu/topics/circular-economy/eu-ecolabel-home_en (accessed on 20 December 2022)

10. Green Public Procurement

*"Although GPP is a voluntary instrument, it has a key role to play in the EU's efforts to become a more resource-efficient economy. It can help stimulate a critical mass of demand for more sustainable goods and services which otherwise would be difficult to get onto the market. GPP is therefore a strong stimulus for eco-innovation."*

https://ec.europa.eu/environment/gpp/index_en.htm (accessed on 20 December 2022)

11. Life-cycle costing

*"Under the 2014 EU procurement rules a contract must be awarded based on the most economically advantageous tender (MEAT). A number of different approaches are available under this general heading, some of which may be considered appropriate for GPP.*

*Cost or price will form part of the assessment of any procedure, and is usually one of the most influential factors. Costs may be calculated on the basis of a product's life-cycle. But how do you define the cost?"*

*"Life-cycle costing (LCC) means considering all the costs that will be incurred during the lifetime of the product, work or service:*

- *Purchase price and all associated costs (delivery, installation, insurance, etc.)*
- *Operating costs, including energy, fuel and water use, spares, and maintenance*
- *End-of-life costs (such as decommissioning or disposal) or residual value (i.e., revenue from sale of product)"*

https://ec.europa.eu/environment/gpp/lcc.htm (accessed on 20 December 2022)

12. Commission Staff Working Document EU—green public procurement criteria for road transport

*"EU green public procurement (GPP) criteria are designed to make it easier for public authorities to purchase goods, services and works with reduced environmental impacts. The use of the criteria is voluntary. The criteria are formulated in such a way that they can, if deemed appropriate by the individual authority, be (partially or fully) integrated into the authority's tender documents with minimal editing. Before publishing a call for tender, public authorities are advised to check the available offer of the goods, services and works they plan to purchase on the market where they are operating. When a contracting authority intends to use the criteria suggested in this document, it shall do so in a manner which ensures compliance with the requirements of EU public procurement legislation (see, for instance, Articles 42, 43, 67(2) or 68 of Directive 2014/24 and similar provisions in other EU public procurement legislation). Practical reflections on this matter is also provided the 2016 handbook on buying green, available at http://ec.europa.eu/environment/gpp/buying_handbook_en.htm"*

https://circabc.europa.eu/ui/group/6e9b7f79-da96-4a53-956f-e8f62c9d7fed/library/e59b49d5-c46d-4f15-9844-d9f538f6719d/details?download=true (accessed on 20 December 2022)

13. Buying green handbook

*"The Handbook is the European Commission's main guidance document to help public authorities buy goods and services with a lower environmental impact. It is also a useful reference for policy makers and companies responding to green tenders."*

https://ec.europa.eu/environment/gpp/buying_handbook_en.html (accessed on 20 December 2022)

14. European Climate Pact

*"Science tells us we have to act urgently to achieve our Paris Agreement goals, notably to limit global warming to well below 2 °C and pursue efforts to limit such warming to 1.5 °C above 1990 levels.*

- *Raise awareness of climate issues and EU actions*
- *Encourage climate action & catalyse engagement*
- *Connect citizens and organisations that act on climate and help them to learn from each other"*

https://climate.ec.europa.eu/eu-action/european-green-deal/european-climate-pact_en (accessed on 20 December 2022)

15. Paris Agreement

*"The Paris Agreement sets out a global framework to avoid dangerous climate change by limiting global warming to well below 2 °C and pursuing efforts to limit it to 1.5 °C. It also aims to strengthen countries' ability to deal with the impacts of climate change and support them in their efforts."*

https://climate.ec.europa.eu/eu-action/international-action-climate-change/climate-negotiations/paris-agreement_en (accessed on 20 December 2022)

16. Katowice climate package

    *"On mitigation, the Katowice Climate Package provides guidance for the second round of Nationally Determined Contributions (NDCs) that countries will submit by 2025. The guidance describes the contents of and approach to mitigation goals and activities to ensure comparability across NDC. The guidelines also address: mitigation co-benefits; capacity-building support to help developing countries produce their NDCs; a common timeframe for communicating NDCs; negative impacts of response measures on certain countries and sectors; and modalities for the operation and use of a public NDC registry."*

https://unfccc.int/process-and-meetings/the-paris-agreement/the-katowice-climate-package/katowice-climate-package (accessed on 20 December 2022)

17. Global Climate Action Agenda

    *"Outside of the formal intergovernmental negotiations, countries, cities and regions, businesses and civil society members across the world are already taking action for the climate."*

https://climate.ec.europa.eu/eu-action/international-action-climate-change/climate-negotiations/global-climate-action-agenda_en (accessed on 20 December 2022)

18. Covenant of Mayors for Climate & Energy—Eastern Europe

    *"The EU Covenant of Mayors for Climate & Energy brings together thousands of local governments voluntarily committed to implementing EU climate and energy objectives."*

    *"The Covenant of Mayors was launched in 2008 in Europe with the ambition to gather local governments voluntarily committed to achieving and exceeding the EU climate and energy targets."*

    *"Not only did the initiative introduce a first-of-its-kind bottom-up approach to energy and climate action, but its success quickly went beyond expectations."*

    *"The initiative now gathers 9000+ local and regional authorities across 57 countries drawing on the strengths of a worldwide multi-stakeholder movement and the technical and methodological support offered by dedicated offices."*

https://www.eumayors.eu/about/covenant-initiative/origins-and-development.html (accessed on 20 December 2022)

19. 2030 Climate Target Plan

    *"The Commission's proposal to cut greenhouse gas emissions by at least 55% by 2030 sets Europe on a responsible path to becoming climate neutral by 2050"*

    *"Based on a comprehensive impact assessment, the Commission has proposed to increase the EU's ambition on reducing greenhouse gases and set this more ambitious path for the next 10 years. The assessment shows how all sectors of the economy and society can contribute, and sets out the policy actions required to achieve this goal."*

https://climate.ec.europa.eu/eu-action/european-green-deal/2030-climate-target-plan_en (accessed on 20 December 2022)

20. 2050 long-term strategy

    *"The transition to a climate-neutral society is both an urgent challenge and an opportunity to build a better future for all. "*

    *"All parts of society and economic sectors will play a role—from the power sector to industry, mobility, buildings, agriculture and forestry. "*

    *"The EU can lead the way by investing into realistic technological solutions, empowering citizens and aligning action in key areas such as industrial policy, finance and research, while ensuring social fairness for a just transition."*

https://climate.ec.europa.eu/eu-action/climate-strategies-targets/2050-long-term-strategy_en (accessed on 20 December 2022)

21. Communication from the Commission—A Clean Planet for all

    *"A European strategic long-term vision for a prosperous, modern, competitive and climate neutral economy"*

    https://eur-lex.europa.eu/legal-content/EN/TXT/?uri=CELEX:52018DC0773 (accessed on 20 December 2022)

22. Commission guidelines on non-financial reporting

    *"The non-financial reporting Directive (2014/95/EU) requires large public interest entities with over 500 employees (listed companies, banks, and insurance companies) to disclose certain non-financial information. As required by the directive, the Commission has published non-binding guidelines to help companies disclose relevant non-financial information in a more consistent and more comparable manner."*

    https://finance.ec.europa.eu/publications/commission-guidelines-non-financial-reporting_en (accessed on 20 December 2022)

23. Factsheet: Financing Sustainable Growth

    *"Sustainable finance makes sustainability considerations part of financial decision-making. This means more climate neutral, energy- and resource-efficient and circular projects. Sustainable finance is needed to implement the Commission's strategy towards achieving the SDGs."*

    https://finance.ec.europa.eu/system/files/2019-06/190618-sustainable-finance-factsheet_en.pdf (accessed on 20 December 2022)

24. Renewed sustainable finance strategy and implementation of the action plan on financing sustainable growth

    *"The recommendations of the* High-level expert group on sustainable finance *form the basis of the* action plan on sustainable finance *adopted by the Commission in March 2018".*

    *"The action plan set out a comprehensive strategy to further connect finance with sustainability."*

    https://finance.ec.europa.eu/publications/renewed-sustainable-finance-strategy-and-implementation-action-plan-financing-sustainable-growth_en (accessed on 20 December 2022)

25. EU taxonomy for sustainable activities

    *"The EU taxonomy is a classification system, establishing a list of environmentally sustainable economic activities. It could play an important role helping the EU scale up sustainable investment and implement the European green deal. The EU taxonomy would provide companies, investors and policymakers with appropriate definitions for which economic activities can be considered environmentally sustainable. In this way, it should create security for investors, protect private investors from greenwashing, help companies to become more climate-friendly, mitigate market fragmentation and help shift investments where they are most needed."*

    https://finance.ec.europa.eu/sustainable-finance/tools-and-standards/eu-taxonomy-sustainable-activities_en (accessed on 20 December 2022)

26. Taxonomy Regulation and delegated acts—Regulation (EU) 2020/852 of the European Parliament and of the Council of 18 June 2020 on the establishment of a framework to facilitate sustainable investment, and amending Regulation (EU) 2019/2088 (Text with EEA relevance)

    *"The Taxonomy Regulation was published in the Official Journal of the European Union on 22 June 2020 and entered into force on 12 July 2020. It establishes the basis for the*

*EU taxonomy by setting out 4 overarching conditions that an economic activity has to meet in order to qualify as environmentally sustainable."*

https://eur-lex.europa.eu/legal-content/EN/TXT/?uri=CELEX:32020R0852 (accessed on 20 December 2022)

27. Sustainable finance package

*"The EU Taxonomy Climate Delegated Act aims to support sustainable investment by making it clearer which economic activities most contribute to meeting the EU's environmental objectives."*

*"On 9 December 2021, a first delegated act on sustainable activities for climate change mitigation and adaptation objectives of the EU Taxonomy ("Climate Delegated Act") was published in the Official Journal. The delegated act is applicable from 1 January 2022."*

https://finance.ec.europa.eu/publications/sustainable-finance-package_en (accessed on 20 December 2022)

28. Transition finance report

*"In January 2021, the European Commission asked the Platform to provide advice on transition financing.1 The Commission identified that more work is needed on how the Taxonomy can enable inclusive transition financing for companies and other economic actors working to improve their environmental impact."*

https://finance.ec.europa.eu/sustainable-finance/tools-and-standards/eu-taxonomy-sustainable-activities_en (accessed on 20 December 2022)

29. Platform on Sustainable Finance

*"The Platform is an advisory body subject to the Commission's horizontal rules for expert groups. Its main purpose is to advise the European Commission on several tasks and topics related to further developing the EU taxonomy and support the Commission in the technical preparation of delegated acts, in order to implement the EU taxonomy."*

https://finance.ec.europa.eu/sustainable-finance/overview-sustainable-finance/platform-sustainable-finance_en (accessed on 20 December 2022)

30. Delegated Act supplementing Article 8 of the Taxonomy Regulation—Commission Delegated Regulation (EU) 2021/2178 of 6 July 2021 supplementing Regulation (EU) 2020/852 of the European Parliament and of the Council by specifying the content and presentation of information to be disclosed by undertakings subject to Articles 19a or 29a of Directive 2013/34/EU concerning environmentally sustainable economic activities, and specifying the methodology to comply with that disclosure obligation (Text with EEA relevance)

*"This delegated act specifies the content, methodology and presentation of information to be disclosed by financial and non-financial undertakings concerning the proportion of environmentally sustainable economic activities in their business, investments or lending activities."*

https://finance.ec.europa.eu/sustainable-finance/tools-and-standards/eu-taxonomy-sustainable-activities_en (accessed on 20 December 2022)

31. EU taxonomy: Complementary Climate Delegated Act to accelerate decarbonisation

*"The Complementary Delegated Act has been published in the Official Journal on 15 July 2022. It will apply from 1 January 2023."*

*"The criteria for the specific gas and nuclear activities are in line with EU climate and environmental objectives and will help accelerating the shift from solid or liquid fossil fuels, including coal, towards a climate-neutral future."*

https://finance.ec.europa.eu/publications/eu-taxonomy-complementary-climate-delegated-act-accelerate-decarbonisation_en (accessed on 20 December 2022)

32. Assessment of nuclear energy—Technical assessment of nuclear energy with respect to the 'do no significant harm' criteria of Regulation (EU) 2020/852 ('Taxonomy Regulation')

    *"Inclusion or exclusion of nuclear energy in the EU taxonomy was a debated subject throughout the negotiations on the Taxonomy Regulation. While there are indirect references in the regulation to the issue of nuclear energy (including on radioactive waste), co-legislators ultimately left the assessment of nuclear energy to the Commission as part of its work on the delegated acts establishing the technical screening criteria."*

    *"The Technical Expert Group on Sustainable Finance (TEG), which was tasked with advising the Commission on the technical screening criteria for the climate change mitigation and adaptation objectives, did not provide a conclusive recommendation on nuclear energy and indicated that a further assessment of the 'do no significant harm' aspects of nuclear energy was necessary."*

    *"As the in-house science and knowledge service of the Commission with extensive technical expertise on nuclear energy and technology, the JRC was invited to carry out such analysis and to draft a technical assessment report on the 'do no significant harm' (DNSH) aspects of nuclear energy including aspects related to the long-term management of high-level radioactive waste and spent nuclear fuel, consistent with the specifications of Articles 17 and 19 of the Taxonomy Regulation."*

    https://finance.ec.europa.eu/system/files/2021-03/210329-jrc-report-nuclear-energy-assessment_en.pdf (accessed on 20 December 2022)

33. Group of Experts on radiation protection and waste management under Article 31 of the Euratom Treaty

    *"A group of independent radiation protection and public health experts is attached to the European Commission to help the EU make decisions concerning radioactivity. Its members are appointed by the Scientific and Technical Committee, referred to in the Euratom Treaty Article 31, and for a duration of 5 years."*

    *"The Commission must consult the group of experts for any updates of the Basic Safety Standards Directive (2013/59/Euratom), which cover safety rules for radiation in applications such as medicine and research."*

    Report: SCHEER review of the JRC report on technical assessment of nuclear energy with respect to the 'do no significant harm' criteria of Regulation (EU) 2020/852 ('Taxonomy Regulation')
    https://finance.ec.europa.eu/document/download/faeb2e5b-aa02-4387-8f33-49930d9dab8d_en?filename=210629-nuclear-energy-jrc-review-scheer-report_en.pdf (accessed on 20 December 2022)
    https://energy.ec.europa.eu/topics/nuclear-energy/radiation-protection/scientific-seminars-and-publications/group-experts_en (accessed on 20 December 2022)

34. Scientific Committee on Health, Environmental and Emerging Risks

    Report: Opinion of the Group of Experts referred to in Article 31 of the Euratom Treaty on Joint Research Centre's report technical assessment of nuclear energy with respect to 'do no significant harm' criteria of Regulation (EU) 2020/852 ('Taxonomy Regulation')
    https://finance.ec.europa.eu/document/download/05796f2d-5deb-4de4-9fb2-da1540c74f9b_en?filename=210630-nuclear-energy-jrc-review-article-31-report_en.pdf (accessed on 20 December 2022)

35. Taxonomy: Final report of the Technical Expert Group on Sustainable Finance

    *"This report sets out the TEG's final recommendations to the European Commission. This report contains recommendations relating to the overarching design of the Taxonomy, as well as guidance on how users of the Taxonomy can develop Taxonomy disclosures. It contains a summary of the economic activities covered by the technical screening criteria."*

    https://finance.ec.europa.eu/system/files/2020-03/200309-sustainable-finance-teg-final-report-taxonomy_en.pdf (accessed on 20 December 2022)

36. Taxonomy Report: Technical Annex

   *"This report represents the overall view of the members of the Technical Expert Group, and although it represents such a consensus, it may not necessarily, on all details, represent the individual views of member institutions or experts. The views reflected in this Report are the views of the experts only. This report does not reflect the views of the European Commission or its services."*

   https://finance.ec.europa.eu/system/files/2020-03/200309-sustainable-finance-teg-final-report-taxonomy-annexes_en.pdf (accessed on 20 December 2022)

37. TEG excel tools to help users of the Taxonomy to implement it in their own activities

   https://finance.ec.europa.eu/document/download/57d4c43c-11d1-42cc-920f-0a3314f7d817_en?filename=sustainable-finance-teg-taxonomy-tools_en.zip (accessed on 20 December 2022)

38. EU Green Bond Standard and labels for green financial products

   *"Once it is adopted by co-legislators, this proposed Regulation will set a gold standard for how companies and public authorities can use green bonds to raise funds on capital markets to finance such ambitious large-scale investments, while meeting tough sustainability requirements and protecting investors."*

   *"This will be useful for both issuers and investors of green bonds. For example, issuers will have a robust tool to demonstrate that they are funding legitimate green projects aligned with the EU taxonomy. And investors buying the bonds will be able to more easily assess, compare and trust that their investments are sustainable, thereby reducing the risks posed by greenwashing."*

   *"The new EUGBS will be open to any issuer of green bonds, including companies, public authorities, and also issuers located outside of the EU."*

   https://finance.ec.europa.eu/sustainable-finance/tools-and-standards/european-green-bond-standard_en (accessed on 20 December 2022)

39. Supervision by the European Securities Markets Authority (ESMA) of reviewers:

   *"External reviewers providing services to issuers of European green bonds must be registered with and supervised by the ESMA. This will ensure the quality of their services and the reliability of their reviews to protect investors and ensure market integrity"*

   https://www.esma.europa.eu (accessed on 20 December 2022)

40. The Sustainable Europe Investment Plan—*What is the Green Deal Investment Plan?*

   *"The European Green Deal Investment Plan (EGDIP), also referred to as Sustainable Europe Investment Plan (SEIP), is the investment pillar of the Green Deal. To achieve the goals set by the European Green Deal, the Plan will mobilise at least €1 trillion in sustainable investments over the next decade. Part of the plan, the Just Transition Mechanism, will be targeted to a fair and just green transition. It will mobilise at least €100 billion in investments over the period 2021–2027 to support workers and citizens of the regions most impacted by the transition."*

   https://ec.europa.eu/commission/presscorner/detail/en/qanda_20_24 (accessed on 20 December 2022)

41. The InvestEU Programme (2021-2027)

   *"The InvestEU Programme builds on the successful model of the Investment Plan for Europe, the Juncker Plan. It will bring together, under one roof, the European Fund for Strategic Investments and 13 EU financial instruments currently available. Triggering at least €650 billion in additional investment, the Programme aims to give an additional boost to investment, innovation and job creation in Europe."*

https://wayback.archive-it.org/12090/20191231194920/https://ec.europa.eu/commissio
n/priorities/jobs-growth-and-investment/investment-plan-europe-juncker-plan/whats-
next-investeu-programme-2021-2027_en (accessed on 20 December 2022)

42. Sustainable finance—obligation for investment firms to advise clients on social and environmental aspects of financial products

*"The EU's action plan on sustainable finance seeks to clarify the duties of investment firms to provide their clients with clear advice on the social and environmental risks and opportunities attached to their investments. This initiative aims to:*

- *shift capital flows away from activities that have negative social and environmental consequences*
- *better assess and manage financial risks resulting from e.g., climate change and environmental damage*
- *direct finance towards economic activities that have genuine long-term benefits for society."*

https://ec.europa.eu/info/law/better-regulation/have-your-say/initiatives/12068-Sust
ainable-finance-obligation-for-investment-firms-to-advise-clients-on-social-and-environme
ntal-aspects-of-financial-products_en (accessed on 20 December 2022)

43. EU labels for benchmarks (climate, ESG) and benchmarks' ESG disclosures

*"Make benchmark methodologies more transparent when it comes to ESG & put forward standards for the methodology of low-carbon and ESG benchmarks in EU."*

https://finance.ec.europa.eu/sustainable-finance/disclosures/eu-labels-benchmarks-clim
ate-esg-and-benchmarks-esg-disclosures_en (accessed on 20 December 2022)

44. Regulation (EU) 2019/2089 of the European Parliament and of the Council of 27 November 2019 amending Regulation (EU) 2016/1011 as regards EU Climate Transition Benchmarks, EU Paris-aligned Benchmarks and sustainability-related disclosures for benchmarks (Text with EEA relevance)

https://eur-lex.europa.eu/legal-content/EN/TXT/?uri=CELEX:32019R2089 (accessed on 20 December 2022)

45. Teg Interim Report on Climate Benchmarks and Benchmarks' ESG Disclosures

*"The main objectives of the new climate benchmarks are to (i) allow a significant level of comparability of climate benchmarks methodologies while leaving benchmarks' administrators with an important level of flexibility in designing their methodology; (ii) provide investors with an appropriate tool that is aligned with their investment strategy; (iii) increase transparency on investors' impact, specifically with regard to climate change and the energy transition; and (iv) disincentivize greenwashing."*

https://finance.ec.europa.eu/document/download/c7c4a69a-27e5-423e-92f3-c746982b38
d7_en?filename=190618-sustainable-finance-teg-report-climate-benchmarks-and-disclosur
es_en.pdf (accessed on 20 December 2022)

46. Handbook of Climate Transition Benchmarks, Paris-Aligned Benchmark and Benchmarks' ESG Disclosures

*"This Handbook is a response to frequently asked questions, which the TEG benchmarks subgroup members encountered when presenting the EU Climate Transition Benchmark (EU CTB), the EU Paris Aligned Benchmark (EU PAB), and the benchmarks' disclosure guidance on environmental, social or governance (ESG) issues. The Handbook commences by (i) clarifying the 7% Reduction Trajectory and (ii) matters of terminology. It continues by explaining (iii) the anti-greenwashing measures, (iv) data sources and estimation techniques as well as (v) related classification. Finally, (vi) ESG disclosure matters are discussed and (vii) further aspects are highlighted. Detailed appendices provide computation and sector mapping guidance."*

https://finance.ec.europa.eu/system/files/2019-12/192020-sustainable-finance-teg-benchmarks-handbook_en_0.pdf (accessed on 20 December 2022)

47. Commission Delegated Regulation (EU) 2022/804 of 16 February 2022 supplementing Regulation (EU) 2016/1011 of the European Parliament and of the Council by specifying rules of procedure for measures applicable to the supervision by the European Securities Markets Authority of certain benchmark administrators (Text with EEA relevance)

    https://eur-lex.europa.eu/legal-content/EN/TXT/?uri=CELEX:32022R0804 (accessed on 20 December 2022)

48. Commission Delegated Regulation (EU) 2022/805 of 16 February 2022 supplementing Regulation (EU) 2016/1011 of the European Parliament and of the Council by specifying fees applicable to the supervision by the European Securities Markets Authority of certain benchmark administrators (Text with EEA relevance)

    https://eur-lex.europa.eu/legal-content/EN/TXT/?uri=CELEX:32022R0805 (accessed on 20 December 2022)

49. EU Climate Transition Benchmarks Regulation—Implementing and delegated acts: full list

    https://finance.ec.europa.eu/document/download/09bd5ef1-f285-4c30-b575-ec371d8640a8_en?filename=climate-benchmarks-level-2-measures-full_en.pdf (accessed on 20 December 2022)

50. Final Report—Guidelines on Disclosure Requirements Applicable to Credit Ratings

    *"To strengthen disclosure on how ESG factors are being considered, ESMA updated its Guidelines on disclosure requirements for credit ratings in July 2019 and has started checking how credit rating agencies apply these new guidelines in April 2020. Moreover, in December 2019, the Commission launched a study on sustainability ratings and research that will explore the types of products that are provided in for ratings and market research, the main players, data sourcing, transparency of methodologies and potential shortcomings in the market. The study is expected to be completed by the Summer 2020."*

    https://www.esma.europa.eu/sites/default/files/library/esma33-9-320_final_report_guidelines_on_disclosure_requirements_applicable_to_credit_rating_agencies.pdf (accessed on 20 December 2022)

51. Regulation (EU) 2019/2088 of the European Parliament and of the Council of 27 November 2019 on sustainability-related disclosures in the financial services sector (Text with EEA relevance)

    *"This Regulation aims to reduce information asymmetries in principal-agent relationships with regard to the integration of sustainability risks, the consideration of adverse sustainability impacts, the promotion of environmental or social characteristics, and sustainable investment, by requiring financial market participants and financial advisers to make pre-contractual and ongoing disclosures to end investors when they act as agents of those end investors (principals)."*

    *"This Regulation maintains the requirements for financial market participants and financial advisers to act in the best interest of end investors, including but not limited to, the requirement of conducting adequate due diligence prior to making investments, provided for in Directives 2009/65/EC, 2009/138/EC, 2011/61/EU, 2013/36/EU, 2014/65/EU, (EU) 2016/97, (EU) 2016/2341, and Regulations (EU) No 345/2013 and (EU) No 346/2013, as well as in national law governing personal and individual pension products. In order to comply with their duties under those rules, financial market participants and financial advisers should integrate in their processes, including in their due diligence processes, and should assess on a continuous basis not only all relevant financial risks but*

*also including all relevant sustainability risks that might have a relevant material nega-
tive impact on the financial return of an investment or advice. Therefore, financial market
participants and financial advisers should specify in their policies how they integrate
those risks and publish those policies."*

https://eur-lex.europa.eu/legal-content/EN/TXT/?uri=CELEX:32019R2088 (accessed on
20 December 2022)

52.    Eba Report on Undue Short-Term Pressure from the Financial Sector on Corporations

*"The report includes important information from public sources as background and
context. First, the overview of academic literature illustrates some evidence of the existence
of short-termism in capital markets, while it provides balanced findings with regard to the
relative roles of bank-based, as opposed to stock-based, financial systems in supporting
short-termism. Second, while EU banks have adopted a diversity of business models, they
apply an average 3- to 5-year time horizon for business planning and strategy-setting
purposes. Time horizons are driven by a number of factors that potentially hamper
the adoption of longer term strategies and activities. Furthermore, the traditional time
horizons of European Union (EU) banks seem to not allow long-term and sustainability
challenges, such as climate-related risks, to be fully taken into account and tackled."*

https://www.eba.europa.eu/sites/default/documents/files/document_library/Final%20
EBA%20report%20on%20undue%20short-term%20pressures%20from%20the%20financial
%20sector%20v2_0.pdf (accessed on 20 December 2022)

53.    EIOPA Potential undue short-term pressure from financial markets on corporates:
       Investigation on European insurance and occupational pension sectors

*"Life insurers and pension funds are usually considered long-term investors: based on
their business models, they receive savings from the households with the promise to paying
back earlier in an unexpected event or in a longer term. Predictability of cash flows is key
for pricing and efficiently managing the savings received. This predictability is provided
by making an appropriate selection of risks that are pooled together and applying the big
numbers law to sufficiently large portfolios; such characteristics typically allows these
investors to follow longer term strategies."*

*"Corporates, in general, benefit from the existence of efficient financial markets to cover
their funding needs. Particularly relevant are the investment habits of life insurers and
pension funds that ensure sufficiently deep, liquid and transparent markets for long-dated
financial instruments."*

*"For these reasons, it is key to monitor whether the insurance and institutions for
occupational retirement provision (IORPs) sectors continue to fulfil their alleged roles
as long-term investors and, in the case of deviations, then investigate the reasons for the
deviation and the potential solutions."*

https://www.eiopa.europa.eu/sites/default/files/publications/reports/eiopa-bos-19-537
_report_on_investigation_undue_short_term_pressure.pdf (accessed on 20 December 2022)

54.    ESMA Report Undue short-term pressure on corporations

*"As regards the comment on the potential short-term effect of the use of benchmarks to
measure performance, ESMA notes that there are legitimate investor protection reasons
to assess the performance of asset managers against market benchmarks as it allows
investors to compare the performance of their collective portfolio management options. In
this context, the UCITS KIID requires the disclosure of the reference benchmark as well
as its historical performance to the end investor. ESMA observes that this requirement is
driven by investor protection concerns which outweigh the potential short- term impact."*

https://www.esma.europa.eu/sites/default/files/library/esma30-22-762_report_on_und
ue_short_term_pressure_on_corporations_from_the_financial_sector.pdf (accessed on 20
December 2022)

55. Communication from the Commission to the European Parliament, the European Council, the Council, the European Central Bank, the European Economic and Social Committee and The Committee of the Regions Action Plan: Financing Sustainable Growth

   *"Finance supports the economy by providing funding for economic activities and ultimately jobs and growth. Investment decisions are typically based on several factors, but those related to environmental and social considerations are often not sufficiently taken into account, since such risks are likely to materialize over a longer time horizon. It is important to recognize that taking longer-term sustainability interests into account makes economic sense and does not necessarily lead to lower returns for investors."*

   *"'Sustainable finance' generally refers to the process of taking due account of environmental and social considerations in investment decision-making, leading to increased investments in longer-term and sustainable activities. More specifically, environmental considerations refer to climate change mitigation and adaptation, as well as the environment more broadly and related risks (e.g., natural disasters). Social considerations may refer to issues of inequality, inclusiveness, labor relations, investment in human capital and communities. Environmental and social considerations are often intertwined, as especially climate change can exacerbate existing systems of inequality. The governance of public and private institutions, including management structures, employee relations and executive remuneration, plays a fundamental role in ensuring the inclusion of social and environmental considerations in the decision-making process."*

   *"This Action Plan on sustainable finance is part of broader efforts to connect finance with the specific needs of the European and global economy for the benefit of the planet and our society."*

   https://eur-lex.europa.eu/legal-content/EN/TXT/?uri=CELEX%3A52018DC0097 (accessed on 20 December 2022)

56. Proposal for a Regulation of the European Parliament and of the Council establishing a carbon border adjustment mechanism-COM/2021/564 final

   *"The first option for a CBAM is an import carbon tax, paid by the importer when products enter the EU. The tax would be collected by customs at the border based on a tax reflecting the price of carbon in the Union combined with a default carbon intensity of the products. Importers would have the opportunity to claim a reduction of the CBAM based on their individual carbon footprint and any carbon price paid in the country of production."*

   *"The second option involves the application on imports of a system that replicates the EU ETS regime applicable to domestic production. This option entails—similar to the system of allowances under the EU ETS—the surrendering of certificates ('CBAM certificates') by importers based on embedded emission intensity of the products they import into the Union, and purchased at a price corresponding to that of the EU ETS allowances at any given point in time. These certificates will not be linked to the EU ETS system of allowances but will mirror the price of these allowances to ensure a coherent approach to the pricing under the EU ETS. National climate authorities will administer the sale of the CBAM certificates and importers will submit declarations of verified embedded emissions in the imported products to these authorities tasked with managing the CBAM and surrender a number of CBAM certificates corresponding to the declared emissions. Such declaration and surrendering will occur—similar to that under the EU ETS—at a yearly reconciliation exercise taking place in the year following the year of importation and based on yearly trade import volumes. The carbon emission intensity of products would be based on default values; however, importers would be given the opportunity, at the moment of the yearly reconciliation exercise, to claim a reduction of the CBAM on the basis of their individual emission performance. They would also be entitled to claim a reduction of the CBAM for any carbon price paid in the country of production (which is not rebated or in other way compensated upon export)."*

*"Option 3 operates in the same way as option 2, however the carbon price of imports is based on actual emissions from third country producers rather than on a default value based on EU producers' averages. Under this option, the importer will have to report the actual emissions embedded in the product and surrender a corresponding number of CBAM certificates."*

*"Option 4 would apply in the same way as option 3. It consists of surrendering CBAM certificates on imported products. However, this option considers also a 10 years phasing in period starting in 2026 during which the free allocations of allowances under the EU ETS would be gradually phased out by 10 percentage points each year and the CBAM would be phased in. During this phasing in period, the CBAM would be reduced proportionally to the amount of free allowances distributed in a given sector."*

*"Option 5 is a variant of Option 3 with a scope extended further down in the value chain. Carbon-intensive materials that are part of semi-finished and finished products would be covered along the value chain. For imports, the CBAM would again be based on the actual emissions from third country producers."*

*"Option 6 consists of an excise duty on carbon-intensive materials covering consumption in the Union of both domestic and imported products, besides the continuation of the EU ETS including the free allocation of allowances covering production in the EU."*

https://eur-lex.europa.eu/legal-content/EN/TXT/?uri=CELEX%3A52021PC0564&lang1=EN&from=pl&lang3=choose&lang2=choose&_csrf=c1e45737-e3b4-4b76-86fb-4367596d569d (accessed on 20 December 2022)

57. Guidance Note on approaches to quantify, verify, validate, monitor and report (upstream emission reductions)

*"As part of the EU climate and energy legislation in place to achieve the greenhouse gas (GHG) reduction targets for 2020 the Fuel Quality Directive (FQD)1 obliges fuels suppliers to reduce the greenhouse gas intensity (life cycle greenhouse gas emissions per unit of energy) of the fuel and energy supplied by them by 6% in 2020 compared to a fuel baseline standard of 2010. The rules on calculation methods laid down in Council Directive (EU) 2015/652 (the implementing Council Directive)2 include the possibility to account for upstream emission reductions and to take those emission reductions into account in the compliance assessment of their/that obligation under the FQD."*

*"This document aims to facilitate the implementation by Member States of this legislation by providing non-binding guidance on approaches to quantify, verify, validate, monitor and report upstream emission reductions, as called for in Recital (6) of the implementing Council Directive. It provides practical aspects on certain topics identified in a series of informal discussions3 held with Member States representatives following the adoption of the implementing Council Directive."*

*"No obligation to use UER as a compliance option. The Fuel Quality Directive and Council Directive (EU) 2015/652 foresee several options for suppliers to reduce the GHG intensity of fuels and energy (and thereby comply with the obligation established in Article 7a of the FQD: (a) blend/supply biofuels, (b) supply fuels with lower GHG intensity such as LPG, CNG and H2, (c) provide electricity for road transport or (d) reduce upstream emission. Suppliers can combine these options as appropriate. There is no obligation to use any specific option."*

*"Possibility to use diverse emission schemes for calculating and certifying emission reductions. Recital 3 of the Council Directive states that, "In order to facilitate the claiming of UERs by suppliers, the use of various emission schemes should be allowed for calculating and certifying emission reductions." In this context, a single upstream emission reduction project generating eligible UERs may be considered to constitute a 'scheme'."*

*"Additionality. For emission reductions to be eligible to be claimed as UERs they must be additional to any emissions changes that would have been expected in the most likely counterfactual scenario."*

*"No double counting. Any particular batch of emission reductions from a given project may only be claimed against FQD GHG emission reduction obligations or other emission reductions targets once. These emission reductions cannot be claimed under the Kyoto Protocol's Clean Development Mechanism or the Joint Implementation. Similarly, upstream emission reductions that have been accounted for third party emission reductions schemes shall not be eligible under the FQD."*

*"Dissociation of upstream reduction and fuel supplied on the market. All GHG reduction projects in any country at upstream production and extraction sites of non- biological raw material for the production of fuels for transport supplied for uses covered by the Fuel Quality Directive should be considered as potentially eligible, so long as they are consistent with the definition in Article 2 of the Council Directive."*

https://climate.ec.europa.eu/system/files/2016-11/guidance_note_on_uer_en.pdf (accessed on 20 December 2022)

58.   EBA Roadmap on sustainable finance

*"The European Banking Authority (EBA) published its roadmap outlining the objectives and timeline for delivering mandates and tasks in the area of sustainable finance and environmental, social and governance (ESG) risks. The roadmap explains the EBA's sequenced and comprehensive approach over the next three years to integrate ESG risks considerations in the banking framework and support the EU's efforts to achieve the transition to a more sustainable economy."*

*"Numerous legislative acts and initiatives allocate to the EBA new mandates and tasks in the area of sustainable finance and ESG risks. Most of these mandates and tasks are closely linked to the EBA's broader objective of contributing to the stability, resilience, and orderly functioning of the financial system. These mandates and tasks cover the three pillars of the banking framework, i.e., market discipline, supervision and prudential requirements, as well as other areas related to sustainable finance and the assessment and monitoring of ESG risks."*

*"This roadmap on sustainable finance builds on and replaces the EBA's first action plan on sustainable finance published in December 2019. The roadmap ensures continuity of actions assumed under the previous action plan, while accommodating the necessary adjustments following the market and regulatory developments, including new mandates and new areas of focus."*

*"In the area of transparency and disclosures, the EBA will continue its work related to the development and implementation of institutions' ESG risks and wider sustainability disclosures. Similarly, the EBA will continue its efforts to ensure that ESG factors and risks are adequately integrated in institutions' risk management framework and in their supervision, including through further developments on climate stress tests. In the area of prudential regulation, the EBA has initiated an assessment of whether amendments to the existing prudential treatment of exposures to incorporate environmental and social considerations would be justified. Furthermore, the EBA will contribute to the development of green standards and labels, and measures to address emerging risks in this field, such as greenwashing. Finally, the EBA will be assessing and monitoring developments in sustainable finance and institutions' ESG risk profile, including on the basis of the expected supervisory reporting."*

*"The roadmap was developed based on the current state of the regulatory framework and reflects the EBA's current expectations regarding specific mandates and tasks. However, considering the ongoing regulatory developments, including the review of the banking*

*package (CRR/CRD), the scope and timelines of specific tasks will only be fully known once the legislative processes are finalised."*

https://www.eba.europa.eu/eba-publishes-its-roadmap-sustainable-finance (accessed on 20 December 2022)

59. European Union Emissions Trading System (EU ETS) data from EUTL

*"The EU Emissions Trading System (ETS) is a central instrument of the EU's policy to fight climate change and achieve cost-efficient reductions of greenhouse gas emissions. It is the world's biggest carbon market. Data about the EU emission trading system (ETS). The EU ETS data viewer provides aggregated data on emissions and allowances, by country, sector and year. The data mainly comes from the EU Transaction Log (EUTL). Additional information on auctioning and scope corrections is included."*

https://www.eea.europa.eu/data-and-maps/data/european-union-emissions-trading-scheme-17 (accessed on 20 December 2022)

60. Strategic Research and Innovation Agenda 2021–2027—Clean Hydrogen Joint Undertaking

*"This document represents the Strategic Research and Innovation Agenda (SRIA) 2021-2027 of the Clean Hydrogen Joint Undertaking (hereafter also Clean Hydrogen JU). It covers therefore the duration of Horizon Europe and identifies the key priorities and the essential technologies and innovations required to achieve the objectives of the joint undertaking."*

*"The Clean Hydrogen JU is the continuation of the successful Fuel Cell and Hydrogen Joint Undertakings (FCH JU and FCH 2 JU), under FP7 and Horizon 2020 (H2020) respectively. It is set up in the form of an institutionalised partnership under the Research and Innovation Framework Programme Horizon Europe."*

https://www.clean-hydrogen.europa.eu/system/files/2022-02/Clean%20Hydrogen%20JU%20SRIA%20-%20approved%20by%20GB%20-%20clean%20for%20publication%20%28ID%2013246486%29.pdf (accessed on 20 December 2022)

61. Intelligent transport systems

*"Intelligent Transport Systems (ITS) are vital to increase safety and tackle Europe's growing emission and congestion problems. They can make transport safer, more efficient and more sustainable by applying various information and communication technologies to all modes of passenger and freight transport. Moreover, the integration of existing technologies can create new services. ITS are key to support jobs and growth in the transport sector. But in order to be effective, the roll-out of ITS needs to be coherent and properly coordinated across the EU."*

*"The European Commission is working with Member States, industry and public authorities to find common solutions to the various bottlenecks for deployment. Through financial instruments the European Commission supports innovative projects in ITS and through legislative instruments it ensures that ITS are rolled out consistently."*

*"In the coming years, the digitalisation of transport in general and ITS in particular are expected to take a leap forwards. As part of the Digital Single Market Strategy, the European Commission aims to make more use of ITS solutions to achieve a more efficient management of the transport network for passengers and business. ITS will be used to improve journeys and operations on specific and combined modes of transport. The European Commission also works to set the ground for the next generation of ITS solutions, through the deployment of Cooperative-ITS, paving the way for automation in the transport sector. C-ITS are systems that allow effective data exchange through wireless technologies so that vehicles can connect with each other, with the road infrastructure and with other road users."*

https://transport.ec.europa.eu/transport-themes/intelligent-transport-systems_en (accessed on 20 December 2022)

62. Digital Single Market Strategy

*"Digital technology is changing people's lives. The EU's digital strategy aims to make this transformation work for people and businesses, while helping to achieve its target of a climate-neutral Europe by 2050. The Commission is determined to make this Europe's "Digital Decade". Europe must now strengthen its digital sovereignty and set standards, rather than following those of others—with a clear focus on data, technology, and infrastructure."*

https://commission.europa.eu/strategy-and-policy/priorities-2019-2024/europe-fit-digital-age_en (accessed on 20 December 2022)

63. Horizon Europe

*"Horizon Europe—#HorizonEU—is the European Union's flagship Research and Innovation programme, part of the EU-long-term Multiannual Financial Framework (MFF) with a budget of €95,5bn (including €75,9bn from the MFF and €5bn from the Next Generation Europe) to spend over a seven-year period (2021–2027)."*

https://www.horizon-eu.eu (accessed on 20 December 2022)

64. Recovery and Resilience Facility

*"As part of a wide-ranging response, the aim of the Recovery and Resilience Facility is to mitigate the economic and social impact of the coronavirus pandemic and make European economies and societies more sustainable, resilient and better prepared for the challenges and opportunities of the green and digital transitions."*

*"The Facility is a temporary recovery instrument. It allows the Commission to raise funds to help Member States implement reforms and investments that are in line with the EU's priorities and that address the challenges identified in country-specific recommendations under the European Semester framework of economic and social policy coordination. It makes available €723.8 billion (in current prices) in loans (€385.8 billion) and grants (€338 billion) for that purpose"*

*"The RRF helps the EU achieve its target of climate neutrality by 2050 and sets Europe on a path of digital transition, creating jobs and spurring growth in the process."*

https://commission.europa.eu/business-economy-euro/economic-recovery/recovery-and-resilience-facility_en (accessed on 20 December 2022)

65. Strategic Plans 2020-2024

*"The purpose of the strategic plans and management plans is to help Commission departments align their work with the Commission's overall policy objectives, and plan and manage activities in order to make the most efficient use of resources."*

*"In their strategic plans, Commission departments describe how they will contribute to the 6 political priorities of the Commission. They define specific objectives for their department for a five-year period as well as indicators to help them track progress. All departments report on progress each year in their annual activity reports."*

https://commission.europa.eu/publications/strategic-plans-2020-2024_en (accessed on 20 December 2022)

66. Regulation (EU) No 1315/2013 of the European Parliament and of the Council of 11 December 2013 on Union guidelines for the development of the trans-European transport network and repealing Decision No 661/2010/EU Text with EEA relevance

*"This Regulation applies to the trans-European transport network as shown on the maps contained in Annex I. The trans-European transport network comprises transport infrastructure and telematic applications as well as measures promoting the efficient management and use of such infrastructure and permitting the establishment and operation of sustainable and efficient transport services."*

*"The infrastructure of the trans-European transport network consists of the infrastructure for railway transport, inland waterway transport, road transport, maritime transport, air transport and multimodal transport, as determined in the relevant sections of Chapter II."*

https://eur-lex.europa.eu/legal-content/EN/TXT/?uri=CELEX%3A32013R1315 (accessed on 20 December 2022)

67. Fit for 55

*"The Fit for 55 package is a set of proposals to revise and update EU legislation and to put in place new initiatives with the aim of ensuring that EU policies are in line with the climate goals agreed by the Council and the European Parliament."*

*"The package of proposals aims at providing a coherent and balanced framework for reaching the EU's climate objectives, which:*

- *ensures a just and socially fair transition*
- *maintains and strengthens innovation and competitiveness of EU industry while ensuring a level playing field vis-à-vis third country economic operators*
- *underpins the EU's position as leading the way in the global fight against climate change"*

https://www.consilium.europa.eu/en/policies/green-deal/fit-for-55-the-eu-plan-for-a-green-transition/ (accessed on 20 December 2022)

68. COM/2020/301—A hydrogen strategy for a climate neutral Europe

*"There are many reasons why hydrogen is a key priority to achieve the European Green Deal and Europe's clean energy transition. Renewable electricity is expected to decarbonize a large share of the EU energy consumption by 2050, but not all of it. Hydrogen has a strong potential to bridge some of this gap, as a vector for renewable energy storage, alongside batteries, and transport, ensuring back up for seasonal variations and connecting production locations to more distant demand centers. In its strategic vision for a climate-neutral EU published in November 2018, the share of hydrogen in Europe's energy mix is projected to grow from the current less than 2% to 13–14% by 2050"*

*"In the integrated energy system of the future hydrogen will play a role, alongside renewable electrification and a more efficient and circular use of resources. Large-scale deployment of clean hydrogen at a fast pace is key for the EU to achieve a higher climate ambition, reducing greenhouse gas emissions by minimum 50% and towards 55% by 2030, in a cost effective way."*

*"Investment in hydrogen will foster sustainable growth and jobs, which will be critical in the context of recovery from the COVID-19 crisis. The Commission's recovery plan highlights the need to unlock investment in key clean technologies and value chains. It stresses clean hydrogen as one of the essential areas to address in the context of the energy transition, and mentions a number of possible avenues to support it."*

*"Moreover, Europe is highly competitive in clean hydrogen technologies manufacturing and is well positioned to benefit from a global development of clean hydrogen as an energy carrier. Cumulative investments in renewable hydrogen in Europe could be up to EUR 180–470 billion by 2050, and in the range of €3-18 billion for low-carbon fossil-based hydrogen. Combined with EU's leadership in renewables technologies, the emergence of a hydrogen value chain serving a multitude of industrial sectors and other end uses could employ up to 1 million people, directly or indirectly. Analysts estimate that clean hydrogen could meet 24% of energy world demand by 2050, with annual sales in the range of €630 billion."*

https://eur-lex.europa.eu/legal-content/EN/TXT/?uri=CELEX:52020DC0301 (accessed on 20 December 2022)

69. COM/2021/557 Amendment to the Renewable Energy Directive to implement the ambition of the new 2030 climate target

*"The European Green Deal (EGD) establishes the objective of becoming climate neutral in 2050 in a manner that contributes to the European economy, growth and jobs. This objective requires a greenhouse as emissions reduction of 55% by 2030 as confirmed by the European Council in December 2020. This in turn requires significantly higher shares of renewable energy sources in an integrated energy system. The current EU target of at least 32% renewable energy by 2030, set in the Renewable Energy Directive (REDII), is not sufficient and needs to be increased to 38–40%, according to the Climate Target Plan (CTP). At the same time, new accompanying measures in different sectors in line with the Energy System Integration, the Hydrogen, the Offshore Renewable Energy and the Biodiversity Strategies are required to achieve this increased target."*

*"The overall objectives of the revision of REDII are to achieve an increase in the use of energy from renewable sources by 2030, to foster better energy system integration and to contribute to climate and environmental objectives including the protection of biodiversity, thereby addressing the intergenerational concerns associated with global warming and biodiversity loss. This revision of REDII is essential to achieve the increased climate target as well as to protect our environment and health, reduce our energy dependency, and contribute to the EU's technological and industrial leadership along with the creation of jobs and economic growth."*

https://eur-lex.europa.eu/legal-content/EN/TXT/?uri=CELEX%3A52021PC0557 (accessed on 20 December 2022)

70. COM/2021/561 Regulation of The European Parliament and of the Council on ensuring a level playing field for sustainable air transport

*"Air connectivity is an essential driver of mobility for EU citizens, of development for EU regions and of growth for the economy as a whole. High levels of air connectivity within the EU, as well as to and from the EU, are best ensured when the EU air transport market functions as a level playing field, where all market actors can operate based on equal opportunities. When occurring, market distortions risk putting aircraft operators or airports at disadvantage towards competitors. In turn, this can result in a loss of competitiveness of the industry, and a loss of air connectivity for citizens and businesses."*

*"In particular, it is essential to ensure a level playing field across the EU air transport market, when it comes to the use of aviation fuel. Indeed, aviation fuel accounts for a substantial share of aircraft operators' costs, i.e., up to 25% of operational costs. Variations in the price of aviation fuel can have important impacts on aircraft operators' economic performance. Furthermore, differences in the price of aviation fuel between geographic locations, as is currently the case between EU airports or between EU and non-EU airports, can lead aircraft operators to adapt their refuelling strategies for economic reasons."*

*"Practices such as 'fuel tankering' occur when aircraft operators uplift more aviation fuel than necessary at a given airport, with the aim to avoid refuelling partially or fully at a destination airport where aviation fuel is more expensive. Fuel tankering leads to higher fuel burn than necessary, hence higher emissions, and undermines fair competition in the Union air transport market. Besides being contrary to the Union's efforts to decarbonise aviation, fuel tankering is also detrimental to healthy competition between aviation market players. With the introduction and the ramp-up of sustainable aviation fuels at Union airports, practices of fuel tankering may be exacerbated as a result of increased aviation fuel costs."*

*"In respect to fuel tankering, the present Regulation therefore aims restore and preserve a level playing field in the air transport sector, while at the same time avoiding any adverse environmental effect."*

*"The Commission adopted in December 2020 the Sustainable and Smart Mobility Strategy. This strategy sets out the objective to boost the uptake of sustainable aviation fuels. Sustainable aviation fuels have the potential to deliver a major contribution to achieving the increased EU climate target for 2030 and the EU's climate neutrality objective. For*

*the purpose of this initiative, sustainable aviation fuels means liquid drop-in fuels substitutable to conventional aviation fuel. In order to decrease significantly its emissions, the aviation sector needs to reduce its current exclusive reliance on fossil jet fuel and accelerate its transition to innovative and sustainable types of fuels and technologies. While alternative propulsion technologies for aircraft such as powered by electricity or hydrogen are making promising advances, their introduction to commercial use will take a considerable effort and time to prepare. Because air transport needs to address its carbon footprint on all flight ranges already by 2030, the role of sustainable aviation liquid fuels will be essential. For this reason, measures are also needed to increase the supply and use of sustainable aviation fuels at Union airports."*

https://eur-lex.europa.eu/legal-content/EN/TXT/?uri=CELEX%3A52021PC0561 (accessed on 20 December 2022)

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
