# Peer review of "The Smart Community: Strategy Layers for a New Sustainable Continental Framework"

_smartcities, doi:10.3390/smartcities6010020_

Round 1

Reviewer 1 Report

The paper deals with an interesting topic but at present has several limitations.

First of all, the study claims to use a quantitative study method (with analysis of the correlation of documents) integrated with a qualitative analysis of the literature.

The study does not explain the implementation of the correlation analysis, the observed variables: it is a strong limitation of the study that does not allow us to understand the goodness of the observed results.

Moreover, it is recommended to integrate the methodological part with studies on mixed methods, given that the study proposes a qualitative-quantitative study.

Once the results have been revised accordingly, the discussion should be better supported as well as the conclusions.

Finally, the limitations of the study need to be clarified.

Author Response

Firstly, we would like to thank to the reviewer for their valuable insights which of course we would integrate in the updated version of the article. We also consider hydrogen fuel as interesting topic and we feel encouraged by such reviews to pursue the research in this area.

In order to add context to our results / findings we added data extracted from our research on the EU documents (kindly see in blue color in the updated version of the article) and also added correlation data on the results sections.

The limitations of the study were added in blue color as part of the results section – more precisely subsection 3.8.

Please see the attachment, for a full revision including the reviewer recommendations. 

Reviewer 2 Report

1.      What is the main question addressed by the research?

The topic investigated in this article is a comparation, contrast and integration effort of European strategies for sustainable development with the evolving market initiatives that are beginning to fuel the fourth industrial revolution. The intention is clear and has been implemented. Nevertheless, the authors should also emphasize the relationship between the presented subject matter and the focus of the journal. In its present form, it is very loose. In addition, it is worth noting the authors' contribution to the development of research on smart cities.

2.      Do you consider the topic original or relevant in the field? Does it
address a specific gap in the field?

The topic of the review of 70 EU documents related to sustainable development and Industry 4.0 is undoubtedly an original idea. It provides knowledge about the coherence of these documents and directions for shaping the EU's future. I have not encountered such considerations so far, so they fill a certain research gap of an administrative and legal nature.

3.      What does it add to the subject area compared with other published
material?

The review can be used both as a synthetic material for further analysis and as an inspiration for empirical research. Its main advantage is the condensation of the most important content of EU documents in the field of sustainable development and the fourth industrial revolution. Usually considerations in this area are fragmentary.

4.      What specific improvements should the authors consider regarding the
methodology? What further controls should be considered?

The methodology is correct and clear. There is no need to improve it. However, the authors could add in the introduction a motivation to undertake the review.

5.      Are the conclusions consistent with the evidence and arguments presented
and do they address the main question posed?

The conclusions are correct, but they mainly refer to the results of the review. Authors should also propose their own recommendations and opinions.

6.      Are the references appropriate?

Yes.

7.      Please include any additional comments on the tables and figures.

The figures are correct.

Author Response

  1. What is the main question addressed by the research?

The topic investigated in this article is a comparation, contrast and integration effort of European strategies for sustainable development with the evolving market initiatives that are beginning to fuel the fourth industrial revolution. The intention is clear and has been implemented. Nevertheless, the authors should also emphasize the relationship between the presented subject matter and the focus of the journal. In its present form, it is very loose. In addition, it is worth noting the authors' contribution to the development of research on smart cities.

Thank you for your comment and the good words you address us. We decided to send our article for publication to the MDPI Smart Cities Journal Special Issue on Accelerating Innovation considering that the title itself stands as an enabler of the Smart Cities framework. We strongly believe that technology development will help not only cities to became smart but all types of communities all the way up to states.

  1. Do you consider the topic original or relevant in the field? Does it
    address a specific gap in the field?

The topic of the review of 70 EU documents related to sustainable development and Industry 4.0 is undoubtedly an original idea. It provides knowledge about the coherence of these documents and directions for shaping the EU's future. I have not encountered such considerations so far, so they fill a certain research gap of an administrative and legal nature.

Thank you very much!

  1. What does it add to the subject area compared with other published
    material?

The review can be used both as a synthetic material for further analysis and as an inspiration for empirical research. Its main advantage is the condensation of the most important content of EU documents in the field of sustainable development and the fourth industrial revolution. Usually considerations in this area are fragmentary.

Thank you!

  1. What specific improvements should the authors consider regarding the
    methodology? What further controls should be considered?

The methodology is correct and clear. There is no need to improve it. However, the authors could add in the introduction a motivation to undertake the review.

Added in blue colour at the very end of the introduction section. Thank you for your suggestion.

  1. Are the conclusions consistent with the evidence and arguments presented
    and do they address the main question posed?

The conclusions are correct, but they mainly refer to the results of the review. Authors should also propose their own recommendations and opinions.

Added in blue colour at the very end of the introduction section. Thank you for your suggestion.

  1. Are the references appropriate?

Yes.

  1. Please include any additional comments on the tables and figures.

The figures are correct.

We would like to thank to the reviewer for his/her comments on the article.

Moreover, please see the attachment for a full revision of the material, according to reviewer suggestions. 

Round 2

Reviewer 1 Report

We thank the authors for the modifications that have improved the quality of the work.